# A Long Term (2005 - 2019) Eddy Covariance Data Set of $CO_2$ and $H_2O$ Fluxes from the Tibetan Alpine Steppe

Felix Nieberding[1,2], Christian Wille[2], Gerardo Fratini[3], Magnus O. Asmussen[1], Yuyang Wang[4,5,6], Yaoming Ma[4,5,6], and Torsten Sachs[2,1]

[1]Technische Universität Braunschweig, Germany
[2]GFZ German Research Centre for Geosciences, Potsdam, Germany
[3]LI-COR Biosciences Inc., Lincoln, Nebraska, USA
[4]Key Laboratory of Tibetan Environment Changes and Land Surface Processes, Institute of Tibetan Plateau Research, Chinese Academy of Sciences, Beijing, China
[5]CAS Center for Excellence in Tibetan Plateau Earth Sciences, Beijing, China
[6]University of Chinese Academy of Sciences, Beijing, China

**Correspondence:** Felix Nieberding (f.nieberding@tu-braunschweig.de), Yaoming Ma (ymma@itpcas.ac.cn)

**Abstract.** The Tibetan alpine steppe ecosystem covers an area of roughly 800,000 km$^2$, containing up to 3.3 % soil organic carbon in the uppermost 30 cm, summing up to 1.93 Pg C for the Tibet Autonomous Region only (472,037 km$^2$). With temperatures rising two to three times faster than the global average, these carbon stocks are at risk of loss due to enhanced soil respiration. The remote location and the harsh environmental conditions on the Tibetan Plateau (TP) make it challenging to derive accurate data on ecosystem-atmosphere exchange of carbon dioxide ($CO_2$) and water vapor ($H_2O$). Here, we provide the first multi-year data set of $CO_2$ and $H_2O$ fluxes from the central Tibetan alpine steppe ecosystem, measured in situ using the eddy covariance technique. The calculated fluxes were rigorously quality checked and carefully corrected for a drift in concentration measurements. The gas analyzer self heating effect during cold conditions was evaluated using the standard correction procedure and newly revised formulations (Burba et al., 2008; Frank and Massman, 2020). A wind field analysis was conducted to identify influences of adjacent buildings on the turbulence regime and to exclude the disturbed fluxes from subsequent computations. The presented $CO_2$ fluxes were additionally gap filled using a standardized approach. The very low net carbon uptake across the 15-year data set highlights the special vulnerability of the Tibetan alpine steppe ecosystem to become a source of $CO_2$ due to global warming. The data is freely available under https://www.doi.org/10.5281/zenodo.3733202 (Nieberding et al., 2020a) and https://doi.org/10.11888/Meteoro.tpdc.270333 (Nieberding et al., 2020b) and may help to better understand the role of the Tibetan alpine steppe in the global carbon-climate feedback.

# 1   Introduction

The Tibetan Plateau is also called "The Third Pole" because it harbors the third largest ice mass on earth, right after the polar regions (Qiu, 2008). It has an area of about 2.5 million $km^2$ at an average elevation of > 4000 m above sea level and includes the entire southwestern Chinese provinces of Tibet and Qinghai, parts of Gansu, Yunnan, Sichuan, as well as parts of India, Nepal, Bhutan and Pakistan. Similarly to the northern high latitudes, the TP is warming considerably faster than the global average, with air temperatures rising at a rate of 0.35 K per decade (from 1970 to 2014) (Yao et al., 2019). At the same time, the TP is experiencing changes in precipitation rates, which alter its water cycle, possibly affecting 1.65 billion people across South East Asia (Cuo and Zhang, 2017). While precipitation is reduced on the southern and eastern margins of the TP, it is enhanced in the central area, partly due to higher temperatures and thus enhanced evaporation, leading to more effective water recycling (Yang et al., 2014; Wang et al., 2018). The majority of the TP is covered by the biggest pastural system in the world, the so called steppe-meadow ecotone, consisting of 450,000 $km^2$ *Kobresia* (syn. *Carex*) *pygmaea* pastures and 800,000 $km^2$ alpine steppe ecosystem (Miehe et al., 2011, 2019). With decreasing precipitation to the west, the *K. pygmaea* pastures are replaced by alpine steppe ecosystems. With 14-48 % mean total vegetation cover, the alpine steppe exhibits considerably less above-ground biomass than the *K. pygmaea* pastures. At least in the eastern part, the alpine steppe soils still contain an almost 30 cm thick, organic-rich layer, which consists of up to 3.3 % soil organic carbon (SOC), summing up to 1.93 Pg C for the Tibet Autonomous Region only (472,037 $km^2$) (Zhou et al., 2019). Hence, the response of $CO_2$ and $H_2O$ fluxes to environmental changes in the Tibetan Plateau grasslands are crucial for the water cycling in greater Asia and the global carbon budget, respectively. While the carbon cycling in *Kobresia* pastures has been studied extensively, the alpine steppe ecosystem remains underrepresented, particularly with regard to long term observations.

This study presents nearly 15 years of eddy covariance data from an alpine steppe ecosystem on the central Tibetan Plateau. The aim of this study is to calculate consistent $CO_2$ fluxes while following standardized quality control methods to allow for comparability between the different years of our record and with other data sets. To ensure meaningful estimates of ecosystem-atmosphere exchange, careful application of the following correction procedures and analyses was necessary: (1) Due to the remote location, continuous maintenance of the eddy covariance (EC) system was not always possible, so that cleaning and calibration of the sensors was performed irregularly. Furthermore, the high proportion of bare soil and high wind speeds led to accumulation of dirt in the measurement path of the infrared gas analyzer (IRGA). The installation of the sensor in such a challenging environment resulted in a considerable drift in $CO_2$ and $H_2O$ gas density measurements. If not accounted for, this concentration bias may distort the estimation of the carbon uptake. We applied a modified drift correction procedure following Fratini et al. (2014) which, instead of a linear interpolation between calibration dates, uses the $CO_2$ concentration measurements from the Mt. Waliguan atmospheric observatory as reference time series. (2) We applied rigorous quality filtering of the calculated fluxes to retain only fluxes which represent actual physical processes. (3) During the long measurement period, there were several buildings constructed in the near vicinity of the EC system. We investigated the influence of these obstacles on the turbulent flow regime to identify fluxes with uncertain land cover contribution and exclude them from subsequent computations. (4) We calculated the de-facto standard correction for instrument surface heating during cold conditions (hereafter called sensor

self heating correction) following Burba et al. (2008) and a revision of the original method following Frank and Massman (2020). (5) Subsequently, we applied the traditional and widely used gap filling procedure following Reichstein et al. (2005) to provide a more complete overview of the annual net ecosystem $CO_2$ exchange. (6) We estimated the flux uncertainty by calculating the random flux error (RE) following Finkelstein and Sims (2001) and by using the standard deviation of the fluxes used for gap filling (NEE_fsd) as a measure for spatial and temporal variation.

## 2  Material and methods

### 2.1  Site description and measurements

The Nam Co Station for Multi-sphere Observation and Research (NAMORS, Chinese Academy of Sciences) is located at 4730 m a.s.l., about 220 km north of the Tibetan capital Lhasa (30° 46' N, 90° 57' S; Fig. 1). It is situated on an almost flat plain between the ENE-WSW oriented Nyainqêntanglha range in 15 km distance to the SSE and lake Nam Co about 1 km to the NW.

The climate at Nam Co is characterized by strong seasonality, with long, cold winters and short but moist summers. The mean annual air temperature measured at the NAMORS research station between 2005 and 2019 was -0.2 °C. During winter, the Westerlies control the general circulation and lead to cold and dry weather, with temperature minima below -20 °C. Although snow storms do occur during winter time, a closed snow cover is seldom reached for longer time periods. In springtime, the TP heats up and allows the melt water to percolate to deeper soil layers. The drought situation increases gradually until the

monsoon rains arrive, typically between May and June. The southern branch of the westerlies needs to shift northward of the Tibetan Plateau so that the humid air masses from the intertropical convergence zone can reach the plateau along meridional river gorges, thus increasing precipitation notably. The annual precipitation ranges from 291 to 568 mm (mean = 403 mm), with the majority occurring during the monsoon season from May to October. During autumn, weather shifts again to clear, cold and dry conditions (Yao et al., 2013). The study site is covered by degraded *Stipa purpurea* alpine steppe vegetation, which includes

species from the families *Artemisia*, *Stipa*, *Poa*, *Festuca* and *Carex* (Li et al., 2018; Miehe et al., 2011). The vegetation heights do not exceed 5 cm due to heavy grazing by yak and sheep and the plant cover is usually less than 50 % (Nölling, 2006). The substrate is mostly soil and loess. The (micro-) meteorological measurements at the NAMORS site were established in 2005 by the Institute of Tibetan Plateau Research (ITP), Chinese Academy of Sciences (CAS) (Ma et al., 2009). The measurement complex is comprised of a 52 m tall Planetary Boundary Layer (PBL) tower measuring air temperature and relative humidity

in 5 different heights and wind speed and wind direction in 3 different heights (1.5 m, 2 m, 4 m, 10 m, 20 m and 1.5 m, 10 m, 20 m, respectively). The 3 m Eddy Covariance measurement tower is equipped with a CSAT3 ultrasonic anomometer (USA) and a Li-7500 open path infrared gas analyzer (IRGA). The separation between the two sensors is 23 cm. In June 2009, the sonic anemometer alignment was changed from 135 degrees to 200 degrees. In 2010, a KH50 krypton hygrometer was installed but the data is not available due to quality constraints. The site is further supplemented with measurements of soil moisture and

soil temperature (0 cm, -10 cm, -20 cm, -40 cm, -80 cm, -160 cm), soil heat flux (-10 cm, -20 cm) and radiation (short and long wave downward and upward radiation, global radiation), precipitation and air pressure measurements. In 2013 the station was

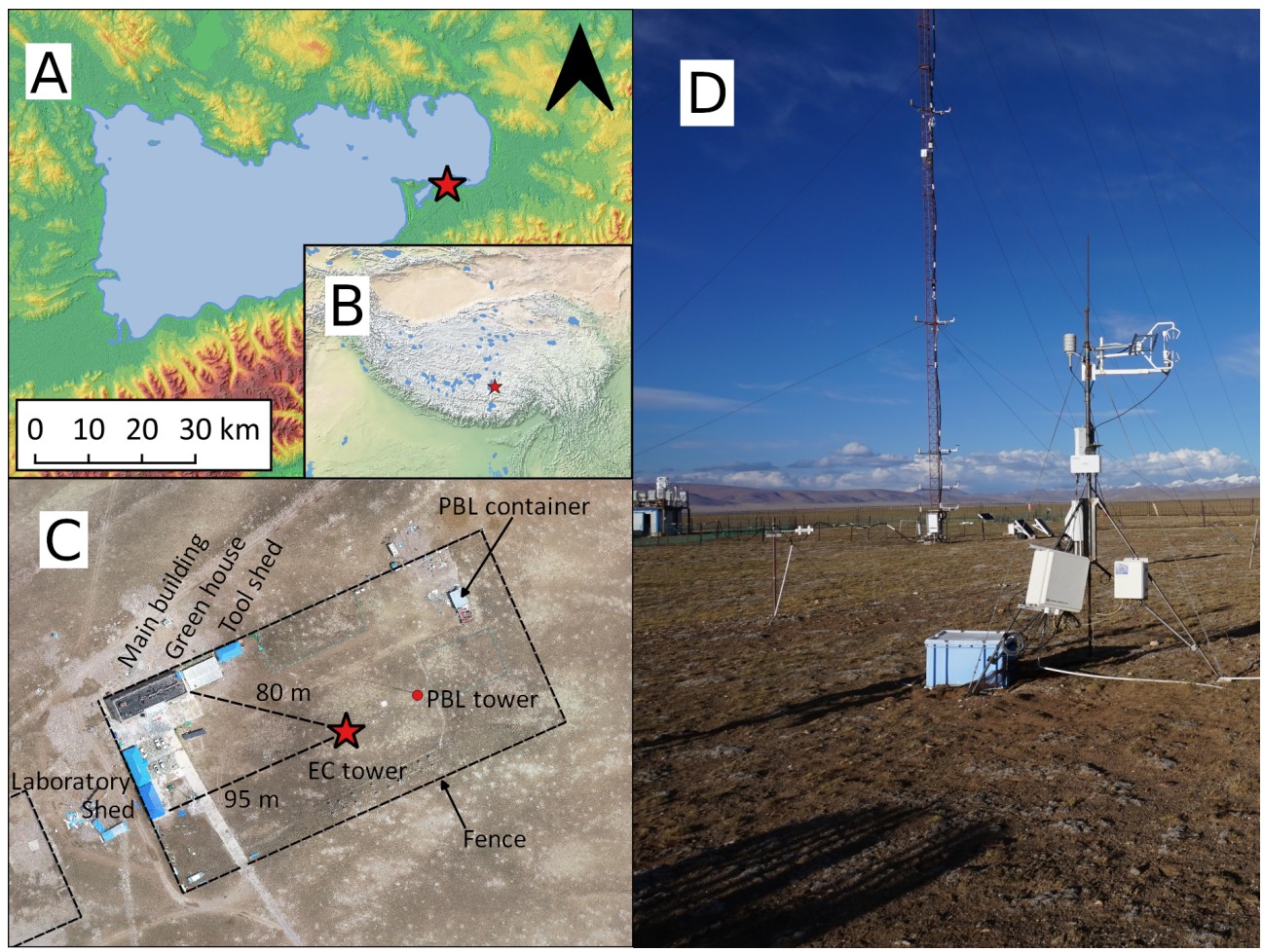

**Figure 1.** (A) Study site at the NAMORS station close to lake Nam Co (ASTER GDEM V3, NASA/METI/AIST/Japan Spacesystems, and U.S./Japan ASTER Science Team, 2019). (B) Overview map showing the location on the Tibetan Plateau, made with Natural Earth. (C) Aerial image of the NAMORS station with the main features in July 2019. Images by Guoshuai Zhang. (D) Photo of the EC system (front), PBL tower and part of the PBL container (back) in May 2019. Photo by Felix Nieberding.

extended for a photosynthetic photon flux density sensor (PPFD) (Zhu et al., 2015). For detailed information about available measurements and sensor types please see Ma et al. (2009).

## 2.2 EC raw data processing

The Eddy Covariance method is a direct micrometeorological approach to estimate turbulent exchange of heat, momentum and matter between the atmosphere and the underlying surface (Aubinet et al., 2012). It is a common approach to estimate the net ecosystem $CO_2$ exchange (NEE), which is the sum of carbon uptake via photosynthesis of green vegetation (Gross

**Table 1.** Flux data processing procedure using EddyPro software (v. 7.0.4, LI-COR Inc.).

| |
|---|
| **Despiking** following Vickers and Mahrt (1997) with the following plausibility ranges: W = 5.0 $\sigma$, $CO_2$ = 3.5 $\sigma$, $H_2O$ = 3.5 $\sigma$, sonic temperature = 3.5 $\sigma$ |
| **Amplitude resolution:** Range of variation = 3.5 $\sigma$, number of bins = 100, accepted empty bins = 70 % |
| **Drop-outs:** extreme bins = 10 percentile, accepted central drop-outs = 10 %, accepted extreme drop-outs = 6 % |
| **Discontinuities** with the following hard flags (hf) and soft flags (sf): U: hf = 4.0, sf = 2.7; W: hf = 2.0, sf = 1.3; Ts: hf = 4.0, sf = 2.7; $CO_2$: hf = 40.0, sf = 27.0; $H_2O$: hf = 40.0, sf = 30.0; variances: hf = 3.0, sf = 2.0 |
| **Skewness** and **kurtosis** with the following hf and sf: Skw limits: hf = ± 2.0, sf = ± 1.0; kur lower limits: hf = 1.0, sf = 2.0; kur upper limits: hf = 8.0, sf = 5.0 |
| **Steadiness of horizontal wind:** Accepted wind relative instationarity = 0.5 |
| **Axis rotation:** Planar fit using 3 wind sectors (80° - 240°, 240° - 320 °, 320° - 80 °) |
| **Detrending** method: Block averaging |
| **Time lags** optimization: Covariance maximization with default ± 1 s |
| Correction for **air density fluctuations**: Application of WPL terms to fluxes (Webb et al., 1980) |
| **Spectral corrections**: Analytic high-pass filtering (Moncrieff et al., 2005) and analytic low-pass filtering (Moncrieff et al., 1997) |

Primary Production, hereafter GPP) and carbon release through autotrophic and heterotrophic respiration (Ecosystem Respiration, hereafter $R_{eco}$). The turbulence measurements were conducted with a CSAT3 ultrasonic anemometer measuring the three-dimensional wind vector. A Li-7500 infrared gas analyzer was placed in close vicinity to the anemometer, measuring the $CO_2$ concentration of the up- and downward moving air parcels. The acquisition frequency is 10 Hz and the fluxes were calculated for every 30-min interval using the raw data processing software EddyPro (v7.0.6, LI-COR Inc.). Table 1 shows the processing and correction procedure which follows standardized and well tested methods including despiking, axis rotation, detrending and data quality flagging based on stationarity, instrument performance, as well as integral turbulence characteristics (Foken and Wichura, 1996; Foken et al., 2004). The data flagging policy is following Mauder and Foken (2006), with "0" for high quality fluxes, "1" for intermediate quality fluxes and "2" for poor quality fluxes. Following the usual meteorological convention, positive values represent fluxes moving away from the surface and negative values represent fluxes moving towards the surface. During 2006 and 2007 the sonic anemometer exhibited a step wise downward vertical tilt of up to 13 degrees. This was accounted for by calculating the planar fit coordinate rotation only for times when the orientation of the sonic anemometer remained constant. The dates were derived visually by analyzing the second rotation angle (pitch), estimated from a preliminary raw data processing using the double rotation method.

## 2.3 Drift correction

The Li-7500 is a so called "dual wavelength, single path" instrument that estimates $CO_2$ and water vapor density ($\rho_c$ and $\rho_v$, respectively) from the amount of radiation passing an ambient air volume in a gas absorbing wavelength, relative to the amount of radiation passing the same sample volume in a non-absorbing reference wavelength. The absorbing wavelengths are 4.25 μm and 2.59 μm for $CO_2$ and $H_2O$, respectively, with both sharing the same non-absorbing reference wavelength of 3.95 μm (Serrano-Ortiz et al., 2008). Absorptance is then converted into an estimate of gas number density by means of an instrument specific curvilinear calibration curve. Due to small sensor specific variations in sources, lens chromatic aberrations, variations in optical filters, detector heterogeneities, and other things, the relationship between absorptance and density is not theoretically predictable but has to be empirically determined for every individual sensor. Following Fratini et al. (2014), every instrument has its own factory derived calibration function ($F$) to describe the exact relationship between absorptance ($a$) and density ($\rho$), depending on air pressure ($Pa$):

$$\rho = PF(\frac{a}{Pa}) \qquad (1)$$

Lens contamination due to mineral dust in the optical path more strongly affects smaller wavelengths, leading to an underestimation of $\rho_c$ and an overestimation of $\rho_v$. These drift errors are usually not accounted for under the assumption, that a slow drift in mean gas concentrations (i.e. over several weeks to months) does not affect the estimation of turbulent fluctuations and, hence, of covariances. This is however not the case. In fact, Serrano-Ortiz et al. (2008) estimated, that a drift in concentration measurements will propagate into an overestimation of the carbon dioxide uptake via the WPL correction. They also showed, that this error is not evenly distributed but has a greater effect during daytime and summer, when sensible heat fluxes are large. Fratini et al. (2014) have shown, that errors in mean concentrations leak into errors in fluxes on account of amplified or dampened estimated fluctuations. Fratini et al. (2014) have also shown that both these effects can be eliminated, possibly completely, when the drift in gas concentration is corrected before raw data processing. Therefore, the offset between measured and reference (i.e. "real") gas concentrations has to be quantified and converted into the corresponding zero offset absorptance biases. Fratini et al. (2014) estimated the reference gas concentrations through linear interpolation of the zero absorptance offset between individual calibration dates, thereby assuming a constant increase of the bias. Due to the remote location on the Tibetan Plateau, user calibrations of the sensor were not performed with due regularity, making this approach not feasible in our case. Instead, we used the $CO_2$ mixing ratio measurements from Mt. Waliguan atmospheric observatory (years 2005-2018, Dlugokencky et al., 2020), situated approximately 1100 km NE of Nam Co, still on the Tibetan Plateau. We averaged the weekly flask measurements to monthly means and fitted the following model to the data:

$$CO_{2\,ref}(t) = p_1 + p_2 * t + p_3 * cos(2 * \pi * t/365) + p_4 * sin(2 * \pi * t/365) + p_5 * cos(4 * \pi * t/365) + p_6 * sin(4 * \pi * t/365), \quad (2)$$

where $t$ is the decimal time in days and $p_i$ are the fit parameters. This model was used to generate the 30-minute $CO_2$ concentration reference time series. The rationale for using this model rather than a linear or spline interpolation was to mimic the general pattern of the atmospheric $CO_2$ background concentration while excluding short term $CO_2$ variations which most likely do not affect the Mt. Waliguan observatory and our site at the same time. We then calculated the $CO_2$ offset used for the drift

correction on a daily basis, as the difference between the daily medians of the measured $CO_2$ concentration and the reference
$CO_2$ concentration. Hence, one offset value was applied to all 30-minute $CO_2$ concentration measurements of each individual
day. In contrast, the $H_2O$ offset was determined as the difference between the 30-minute $H_2O$ concentration measured by the
Li-7500 gas analyzer and the $H_2O$ concentration calculated from auxiliary low frequency measurements of relative humidity,
temperature and air pressure. The time series of 30 minute concentration offset values were imported as dynamic metadata file
in EddyPro. Together with the sensor specific calibration information we can use Eq. (3) (which is Eq. (10) from Fratini et al.,
2014) to calculate the true absorptance ($a$) from measured absorptance ($a_m$) and any absorptance offset ($a_0$), which is then
converted back to densities or mixing ratios.

$$a = \frac{a_m - a_0}{1 - a_0} \tag{3}$$

The corrected 10 Hz concentration measurements are then used to repeat raw data calculation of the fluxes and subsequent
corrections, including application of WPL terms following the methodology in Sect. 2.2. Note that all conversions between
absorptance and number density require the calibration function of the specific instrument.

## 2.4 Quality filtering

The correct application of the eddy covariance method requires a wide range of assumptions and works only within certain
conditions. To ensure meaningful flux calculations, the raw data needs to be tested and flagged very thoroughly. We used the
quality flags and tests implemented in EddyPro and applied additional filtering for low frequency outliers using openeddy R
package from Ladislav Šigut, who implemented the quality control procedure following Mauder et al. (2013). The flagging
scheme remains the same as above with "0" for high quality fluxes, "1" for intermediate quality fluxes and "2" for poor quality
fluxes. As a first step, we manually removed periods with obvious sensor malfunctioning, especially in 2012 and 2018. To
identify periods with insufficient turbulent mixing, we estimated the friction velocity (u*) threshold using the REddyProc R
package. Fluxes with u* < 0.08 m s$^{-1}$ were excluded from subsequent calculations. During the long measuring period, spanning
nearly 15 years, several buildings and scientific infrastructure were constructed in close vicinity of the eddy covariance tower.
During the development of the NAMORS, from the foundation with only a few tents in 2005 to a well-equipped research
station in 2019, we approximated five times with significant changes in constructions. In 2009 the PBL container, the shed and
the solar panel were set up. In 2010 the main building and the green house were constructed. In 2012 the shed was rotated
to become the laboratory and the tool shed next to the greenhouse was added. Finally, in 2019 the garage was relocated and
extended south of the laboratory and the solar panels were removed. To assess possible influences on the flow and turbulence
regime, we analyzed the wind direction distribution of the mean wind speed and the turbulent kinetic energy. We accounted for
possibly disturbed turbulence, by applying the planar fit axis rotation for three different wind sectors during flux calculation
(see Sect. 2.2). Furthermore, we generated a quality flag (qc_wind_dir) indicating whether a flux originates from a disturbed
sector. Table 2 shows the disturbed sectors which were excluded from subsequent calculations. The $CO_2$ and $H_2O$ fluxes and
their respective densities were checked for repeating values as they are a sign of malfunctioning equipment. We furthermore
extracted hard flags of skewness and kurtosis (see Table 1) and combined all flags to a preliminary composite which was used

**Table 2.** Disturbed wind sectors and times.

| From | To | Back of USA | PBL container | Main buildings |
|---|---|---|---|---|
| 2005-12-04 | 2009-06-30 | 305°–325° | - | 260°–280° |
| 2009-06-30 | 2010-01-30 | 10°–30° | 30°–50° | 260°–280° |
| 2010-01-30 | 2011-12-31 | 10°–30° | 30°–50° | 250°–300° |
| 2012-01-01 | 2018-12-31 | 10°–30° | 30°–50° | 250°–315° |
| 2019-01-01 | 2019-09-07 | 10°–30° | 30°–50° | 245°–315° |

as a prerequisite for subsequent low frequency despiking of the flux time series. Please note that the $H_2O$ gas densities and concentrations in the EddyPro full output file are calculated mainly from low frequency measurements of air temperature, pressure and relative humidity, probably because these are deemed more accurate than the high frequency measurements of the IRGA. To enhance comparability we extracted the high frequency $H_2O$ gas densities from the EddyPro statistics output file (variable 'mean(h2o)') and calculated the mole fraction and mixing ratio. These variables were quality filtered following the same scheme as above and are supplied with the data set (suffix: _Li7500). To account for seasonal variations, despiking is done within blocks of 13 consecutive days by comparing each record ($v_i$) with its neighbors via double differencing to produce its score $x$:

$$x = (v_i - v_{i-1}) - (v_{i+1} - v_i) \tag{4}$$

A measurement gets flagged if $x$ is larger or smaller than the median of the scores ($M_x$) $\pm$ the scaled absolute median *MAD*:

$$M_x + \frac{z * MAD}{0.6745} < x < M_x - \frac{z * MAD}{0.6745} \tag{5}$$

with $MAD$ being defined as:

$$MAD = median(x - M_x) \tag{6}$$

The constant 0.6745 in Eq. (5) corresponds to the Gaussian distribution and allows for comparability of MAD with the scaling factor $z$, which determines how rigorous the algorithm screens for outliers (Papale et al., 2006). The lower the value, the stricter the screening, with our setting left to the default $z = 7$. This procedure was repeated iteratively 10 times or until no outliers were detected anymore. For every measurement, the flags were combined to an overall quality flag and fluxes and concentrations with poor quality (flag = "2") were removed from subsequent computations. The quality flagged high frequency molar densities were also incorporated in the composite flag of the $CO_2$ and $H_2O$ fluxes. The R script with the exact calculations can be found in the supplementary material.

## 2.5 Sensor self heating correction

When using an open path IRGA, it is necessary to correct for air density fluctuations caused by fluctuations of temperature and water vapor in the measurement path. The WPL correction compensates for the naturally occurring density fluctuations

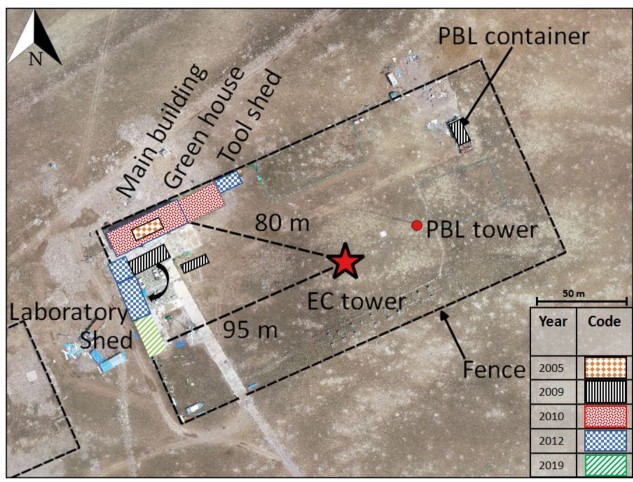

**Figure 2.** Map of the NAMORS station showing the identified changes in construction from 2005 to 2019. Aerial images provided by Guoshuai Zhang.

and should be applied in any case (Webb et al., 1980). Furthermore, especially during cold conditions (low temperatures below -10 °C), an apparent $CO_2$ uptake may be measured, which is caused by conductive, convective, and radiative heat exchange processes happening in the measurement path (Burba et al., 2008). These stem from heating of internal electronics during normal operation, as well as solar radiation encountered by different instrument parts surrounding the open sampling path of the

sensor. This correction is necessary for pre-2010 models of the Li-7500 or for newer instruments (e.g. Li-7500A, Li-7500RS) with summer setting but used in a very cold environment (Oechel et al., 2014). Although the size of the heating correction is quite small (i.e. 10-50 times smaller than the WPL-correction) the small bias can lead to an overestimation of net ecosystem $CO_2$ uptake when integrating over long periods in cold environments (Oechel et al., 2014). Burba et al. (2008) developed a correction procedure which is well tested and fully implemented in the EddyPro software. The procedure depends on a range

of correction factors, which were developed from the original sensor setup in Nebraska, USA. Our study site on the Tibetan Plateau displays different environmental conditions than those in which the correction was tested, namely an inclined IRGA, lower ambient temperatures and strong winds, as well as possible snow and ice deposits on parts of the instrument. In a recent publication, Frank and Massman (2020) tested the correction procedure for a "cold, windy, high-elevation mountainous site" and found inconsistencies in the Burba et al. (2008) correction: (1) The Burba et al. (2008) correction contains boundary layer

adjustment terms for non-flat surfaces but the top- and bottom surfaces of the Li-7500 are flat. This leads to an underestimation of the surface heat fluxes which is an order of magnitude too small. (2) The weightings of the bottom and top surface heat fluxes are "improbable and an order of magnitude too large" (Frank and Massman, 2020). While these errors canceled out during the study of Frank and Massman (2020), this may not be the case for other field sites. Following the recommendations of Frank and Massman (2020), we discarded the boundary layer adjustment terms for non-flat surfaces from the calculations

and applied their newly calculated weightings of the bottom and top surface heat fluxes, thus emphasizing the role of the spar

in self heating. We first reproduced the Burba et al. (2008) correction like it is implemented in EddyPro and then adjusted the equations as described above. Please note that we used the quality filtered variable co2_molar_density from the EddyPro full output file in order to reproduce the calculations.

## 2.6 Gap filling

In order to obtain a $CO_2$ flux time series as complete as possible, we filled the data gaps using the marginal distribution sampling (MDS) algorithm (Falge et al., 2001; Reichstein et al., 2005), implemented in the REddyProc R package by Wutzler et al. (2018). Depending on the length of the data gap and the availability of the meteorological input variables radiation (Rg), air temperature (Tair) and water vapor deficit (VPD), the missing $CO_2$ flux values are derived from a look up table (LUT) or from mean diurnal course (MDC). The LUT approach replaces the missing value with the average value under similar

meteorological conditions within a certain time window. Meteorological conditions are similar if Rg deviates not further than 50 W m$^{-2}$, Tair not further than 2.5 °C and VPD not further than 5.0 hPa. If no similar conditions can be found within an appropriate time window, the missing value is replaced using the average value at the same time of the day (1 hour) (MDC). If the missing value can not be filled during the initial time period (7 - 14 days), the time window size is increased and the procedure repeated until the value can be filled or the data gap gets too long for reliable gap filling (i.e. > 60 days). Prior to the

processing, we excluded the lower and upper 0.2 % of the fluxes and discarded physiological implausible night time fluxes < -0.1 μmol m$^{-2}$ s$^{-1}$. To estimate the uncertainty of the gap filling procedure, we used the method implemented in REddyProc R package, which, besides filling real gaps, creates artificial gaps from otherwise available data and fills them in the same way as if it was a real gap (see section 2.6). The model-value residual should be considered when aggregating the gap-filled time series to daily or annual estimates of NEE, GPP and R$_{eco}$. We included the filled values for the artificially created data gaps,

as well as quality flags for the gap filling procedure, with "0" for measured data, "1" for high reliability, "2" for intermediate reliability and "3" for poor reliability of the gap-filled values. The full MDS algorithm is described in Wutzler et al. (2018) and the R script used in this study can be found in the appendix.

## 2.7 Flux uncertainty estimation

As with all measurements, the reported fluxes are subject to uncertainty, consisting of a systematic and a random part. System-

atic uncertainties may occur e.g. from having an imperfect measurement setup or, like in our case, due to limited maintenance and calibration of the sensors (see section 2.3). We applied a wide range of methods to filter and compensate for systematic errors. Most importantly, we tested for fulfillment of basic EC assumptions using integral turbulence characteristics and steady state test (e.g., Foken and Wichura, 1996) and compensated for air density fluctuations and high- and low-frequency losses (see Sect. 2.2, 2.4 and 2.5). In contrast to systematic uncertainties, random errors do not bias the flux in any direction but

reduce the overall confidence (i.e. precision) of the reported values (Richardson et al., 2012). Random uncertainties mainly arise from the stochastic nature of turbulence, footprint variability, as well as from instrument noise and the resolution at which samples are recorded (Richardson et al., 2012). Hence, it is important to estimate the random uncertainty, especially in places with rather low magnitude of fluxes, as it is in our case. We estimated the random flux error (RE) using the mathematically

rigorous and fully implemented approach by Finkelstein and Sims (2001). This method calculates the uncertainty arising from insufficient sampling of large eddies with high spectral energy, the so-called sampling error. As these large turbulences appear irregularly during sampling, the error is random and can be estimated. First, the so-called Integral Turbulence time-Scale (ITS) is calculated. Basically, the ITS is the covariance between vertical wind velocity and gas concentration as a function of lag time between these two time series (Holl et al., 2019). With increasing time lag, the cross correlation function typically decreases towards values close to zero, indicating an increasing non-correlation of the two time series. In practice, the correlation function must be stopped, otherwise it would go infinitely towards zero. We stopped the integral as soon as the cross-correlation function – which always starts at 1 – crosses the x-axis (i.e. first crossing 0). In case the cross-correlation function would never cross the x axis, a default time value can be provided at which the function is stopped. We set this "maximum correlation period" to 5 s in order to keep computational performance high. Once the ITS is calculated, the RE can be estimated based on the calculation of the "variance of covariance" (Finkelstein and Sims, 2001).

Because the RE method is based on the half hourly auto- and cross-correlation of the vertical wind component and the scalar of interest (e.g. air temperature or gas concentration), it contains only very limited information about spatial, temporal or meteorological variability (El-Madany et al., 2018). During the MDS gap filling procedure (Reichstein et al., 2005), the missing NEE values are (mostly) calculated using a look-up table approach with quite narrow meteorological bins (bin width: $Rg$ = 5 W m$^{-2}$, VPD = 5 hPa, $T_{air}$ = 2.5 °C) within a short time window (+/- 7 days). During these conditions, the vegetation should not change a lot and hence, the ecosystem response to atmospheric drivers should remain the same. Any variability of the flux measurements probably stems from the temporal (+/- 7 days sampling window) and spatial (changes in footprints between 30-min fluxes) heterogeneity at the site. Hence, we could use the standard deviation of the fluxes used for gap filling (NEE_fsd) as an additional measure for flux uncertainty to complement the random uncertainty estimation of Finkelstein and Sims (2001).

# 3 Results

## 3.1 Drift correction

Figure 3 illustrates the effect of the drift correction: Before drift correction, the $CO_2$ mixing ratios were underestimated substantially. Note the rapid divergence from the model after user calibrations were performed in 2009, 2012 and 2019. In June 2017 the original sensor was replaced with another one that was factory calibrated in March 2016. The drift correction eliminates the daily divergence from the modeled background concentration while keeping high frequency fluctuations for computation of the 30-min averaged fluxes. Remember that the drift correction applies an offset to the raw data and is subject to the subsequent flux calculation and correction procedures. Before application of WPL and spectral attenuation terms, the corrected fluxes yielded higher carbon uptake during daytime and summer, than the uncorrected fluxes. If the WPL and spectral attenuation correction is taken into account, the carbon uptake of the drift corrected fluxes gets considerably smaller than the uncorrected fluxes, especially during times with high sensible heat fluxes. The findings are in compliance with instrument theory of operation and the results from numerical simulations and field data analysis as conducted by Fratini et al. (2014) and with Serrano-Ortiz

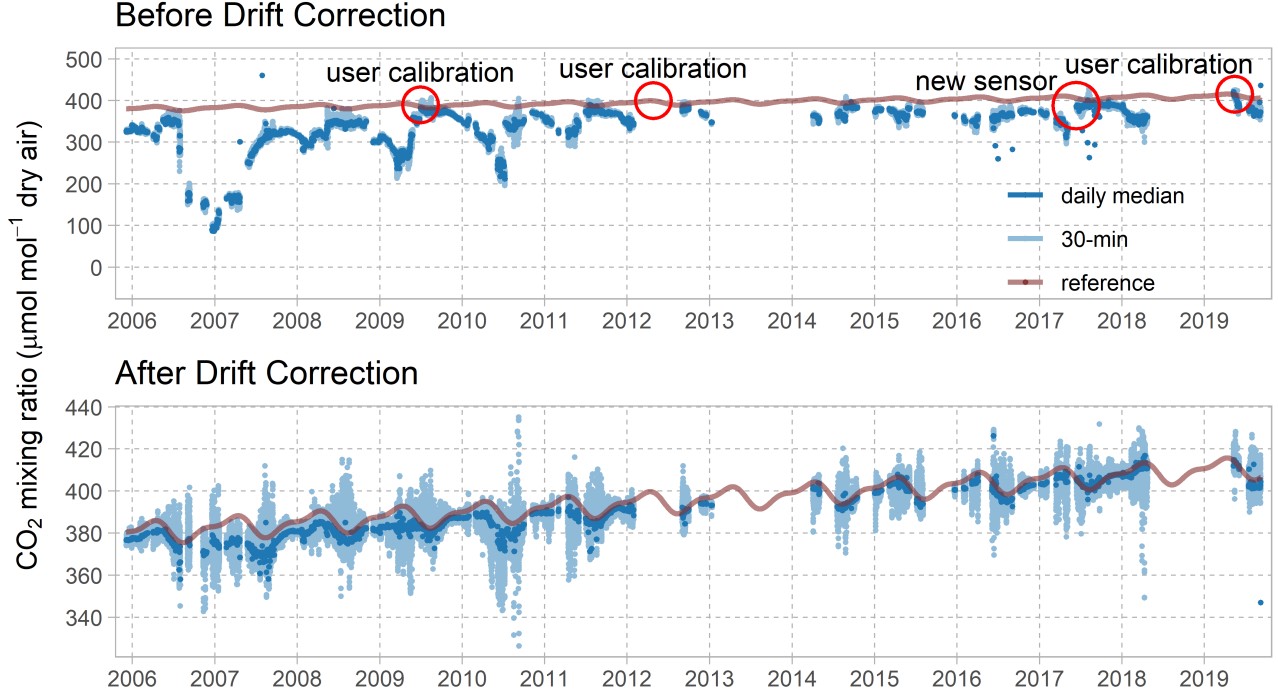

**Figure 3.** Daily median and 30-min $CO_2$ dry air mixing ratios and modeled $CO_2$ background concentration before and after drift correction. $CO_2$ mixing ratios have been checked for repeating values and outliers using the same algorithms as in Sect. 2.4. Please note the different y-axis scales.

et al. (2008), who analyzed the propagation of a systematical underestimation of the $CO_2$ concentration on the $CO_2$ fluxes via the WPL terms. However, the software implementation of the drift correction is still in development and was not yet officially released (i.e. you can not use it via the GUI), so unfortunately it still contains a software error (bug): Every data gap in the $H_2O$

280 reference concentration time series (due to e.g. missing low frequency meteorological data) produces a corrupted $H_2O$ mixing ratio record in the following half hour, which also affects the calculation of the $CO_2$ mixing ratio. This issue was raised with the EddyPro developers and will be fixed in one of the upcoming releases. Because this error does not affect the calculation of the fluxes or other variables, we removed the erroneous values by setting plausibility limits.

### 3.2 Data availability and quality filtering

285 Table A1 lists the yearly $CO_2$ and $H_2O$ flux data availability after raw data processing and application of the various quality flags (wind direction, Ustar and QC, including flux despiking). After raw data processing, more than 50 % of the $CO_2$ fluxes are available for the years 2007 to 2011, as well as for 2017. Two periods with obviously corrupted flux measurements were excluded from the data set right after raw data processing. $CO_2$ and $H_2O$ fluxes and concentrations were discarded from 2012-01-30 02:00 to 2012-08-31 15:00 and from 2018-06-01 00:00 to 2018-06-30 23:30 due to sensor malfunctioning. All times

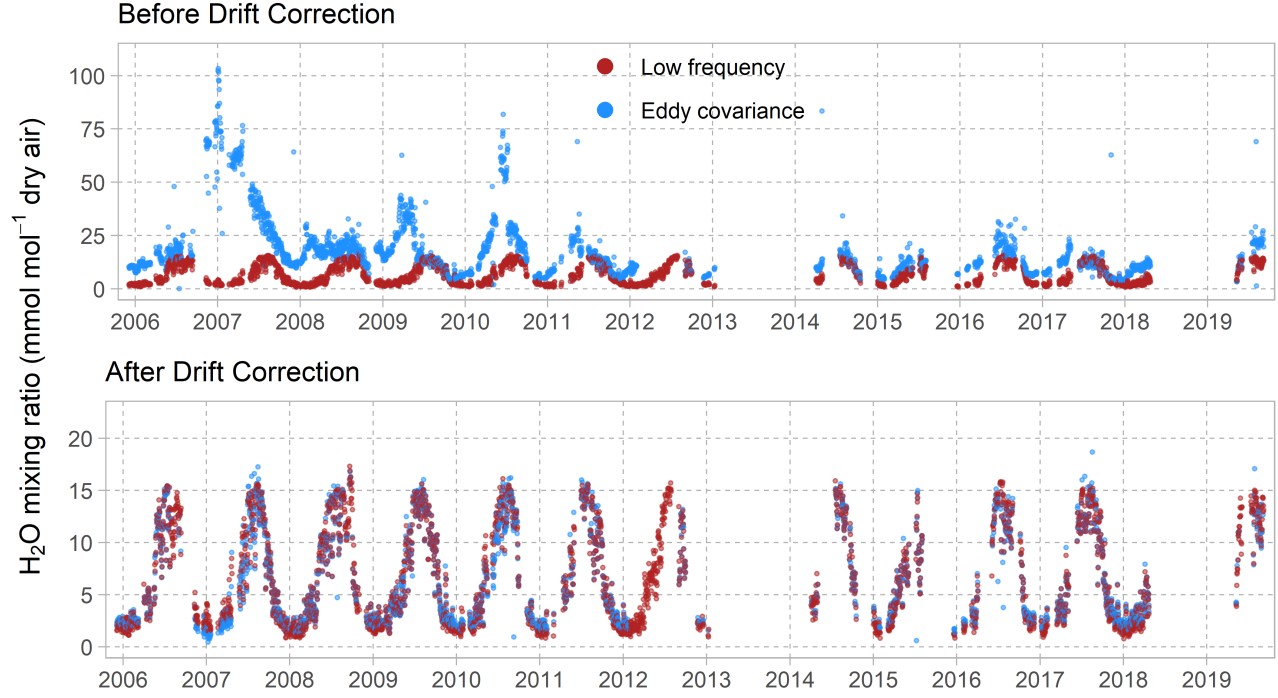

**Figure 4.** Half hourly $H_2O$ dry air mixing ratios and low frequency reference concentration before and after drift correction. $H_2O$ mixing ratios have been checked for repeating values and outliers using the same algorithms as in Sect. 2.4. Please note the different y-axis scales.

are in China Standard Time (CST, UTC+8), which is approximately 2 hours later than local solar time. Figure 5 shows the data availability of the $CO_2$ fluxes throughout the individual years before and after gap filling (see also Sect. 2.6). The overall data availability differs substantially between the years. In 2013 more than 90 % of the data are missing. In 2014 the complete winter period is missing and about one year of data is missing from May 2018 to May 2019. Continuous data gaps of up to six months occur irregularly throughout the whole data set, whereas shorter gaps can be found within periods of continuously available data. The large gaps occurred mainly due to hardware errors and power outages whereas smaller gaps are caused by raw data processing and subsequent filtering of fluxes with poor quality or due to violation of basic EC assumptions (see Sect. 2.2 and 2.4). NEE gap filling increased the overall data availability for $CO_2$ fluxes from 52 % to 72 % in total, with seven complete years. The data set contains quality flags for each flux, indicating whether it was gap filled and how well the gap filling mechanism performed (Reichstein et al., 2005). The quality of the gap filled fluxes was derived by treating individual available values as data gaps and filling them as if they were real data gaps. When aggregating the residuals to an overall error estimate of the model performance, autocorrelation has to be taken into account because no independent training data is available. The model-data residuals were checked for empirical autocorrelation, indicated positive autocorrelation until a lag of up to 65 records, decreasing the number of effective observations from 62465 to 7047. Taking this into account, Pearsons correlation coefficient is 0.91 with a root mean square error (RMSE) of 1.6 μmol m$^{-2}$ s$^{-1}$.

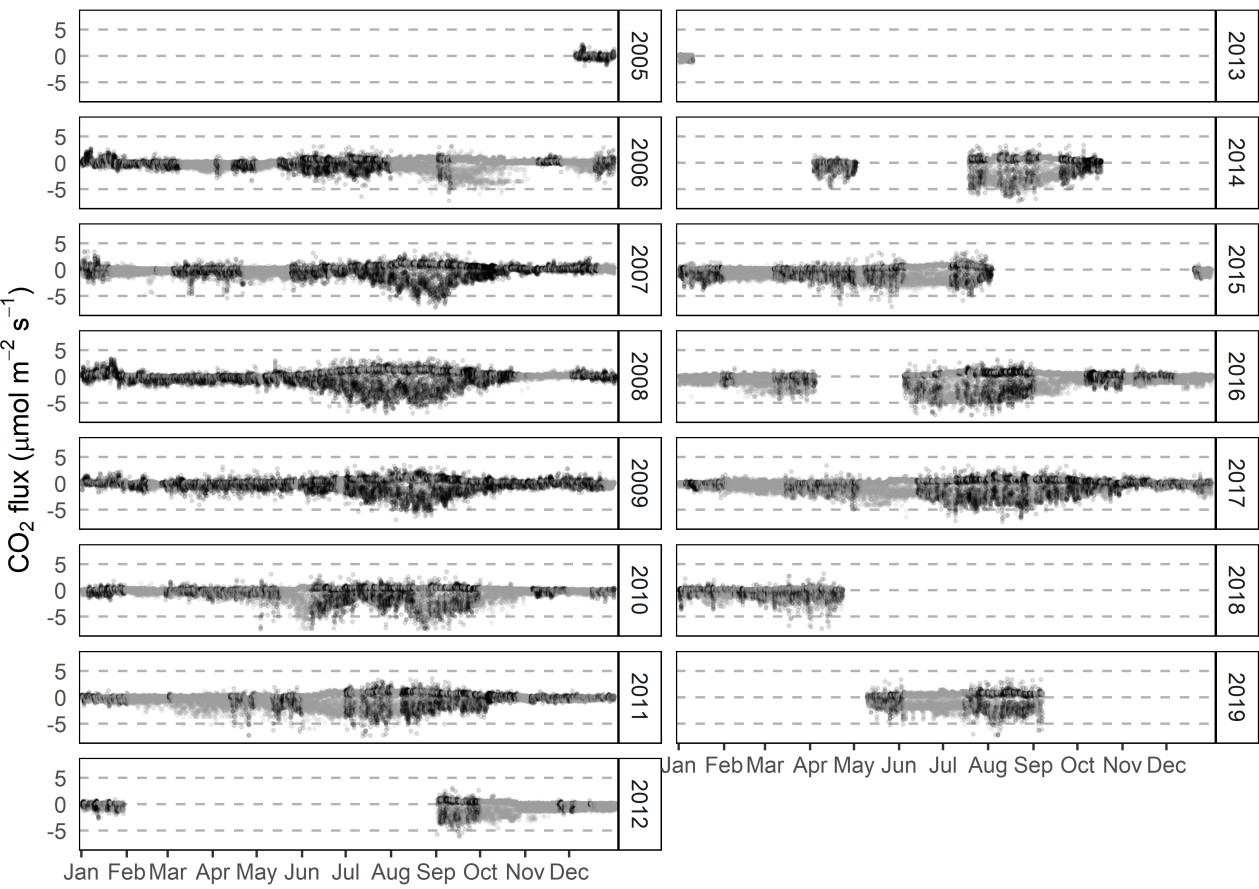

**Figure 5.** Data availability after gap filling using MDS algorithm (Wutzler et al., 2018).

Figure 2 shows how constructions changed in 2009, 2010, 2012 and 2019. It was not possible to assess when exactly the different buildings were constructed, hence we focused on the single most severe change, the set up of the two-storey main building in 2010. To assess the impact on the wind field, we calculated wind rose plots for wind speed and TKE (Fig. 6) before and after 2010. It is to note that the wind regime is superimposed by large and small scale circulation systems. During summer, the Indian and East Asian summer monsoon blow from the southern directions and during winter, the westerlies provide air masses from western directions. Furthermore, the wind may be deflected along the Nyainqêntanglha range and exhibits a diurnal pattern due to a land–lake circulation system, caused by the large water masses of Nam Co (see also Biermann et al., 2014). Nevertheless, the differences before and after 2010 are clearly identifiable. A shift in the wind direction contribution

**Figure 6.** Wind roses showing the wind speed distribution and turbulent kinetic energy (TKE) in 5 ° binned wind directions at NAMORS EC station before and after 2010.

away from the building and towards more western directions can be observed. Furthermore, wind speed and TKE increase substantially in western direction while decreasing in the direction of the main building.

### 3.3 Sensor self heating correction

The monthly mean diurnal course of the three $CO_2$ flux time series in Fig. 7 clearly shows the effect of the sensor self heating correction during cold conditions (air temperature $< 0$ °C). The effect of the correction procedure following Burba et al. (2008) and the revised equations of Frank and Massman (2020) are very similar. We see various problems associated with the SSH corrections: (1) the corrections create strong artifacts during the transition between day and night, (2) SSH-corrected nighttime $CO_2$ fluxes during the cold months are very high — at about the same level as the nighttime $CO_2$ fluxes during summer — suggesting an over-correction of SSH effects, and (3) the winter-time diurnal course of $CO_2$ flux with daytime uptake of $CO_2$ -– which is assumed to be the effect of the SSH -– does not disappear, but the daytime $CO_2$ flux is merely offset by what seems to be a more or less constant flux value. This leads us to the conclusion that the effect of the SSH is very small at our site and that the application of the standard correction (Burba et al., 2008) and its improved version Frank and Massman (2020) lead to an undue over-correction of this effect. To test our conclusion about the SSH effect at our site, we calculated the mean diurnal course of $CO_2$ fluxes during cold periods with a closed snow cover (Fig. A1). Under these (rare) conditions, we expect

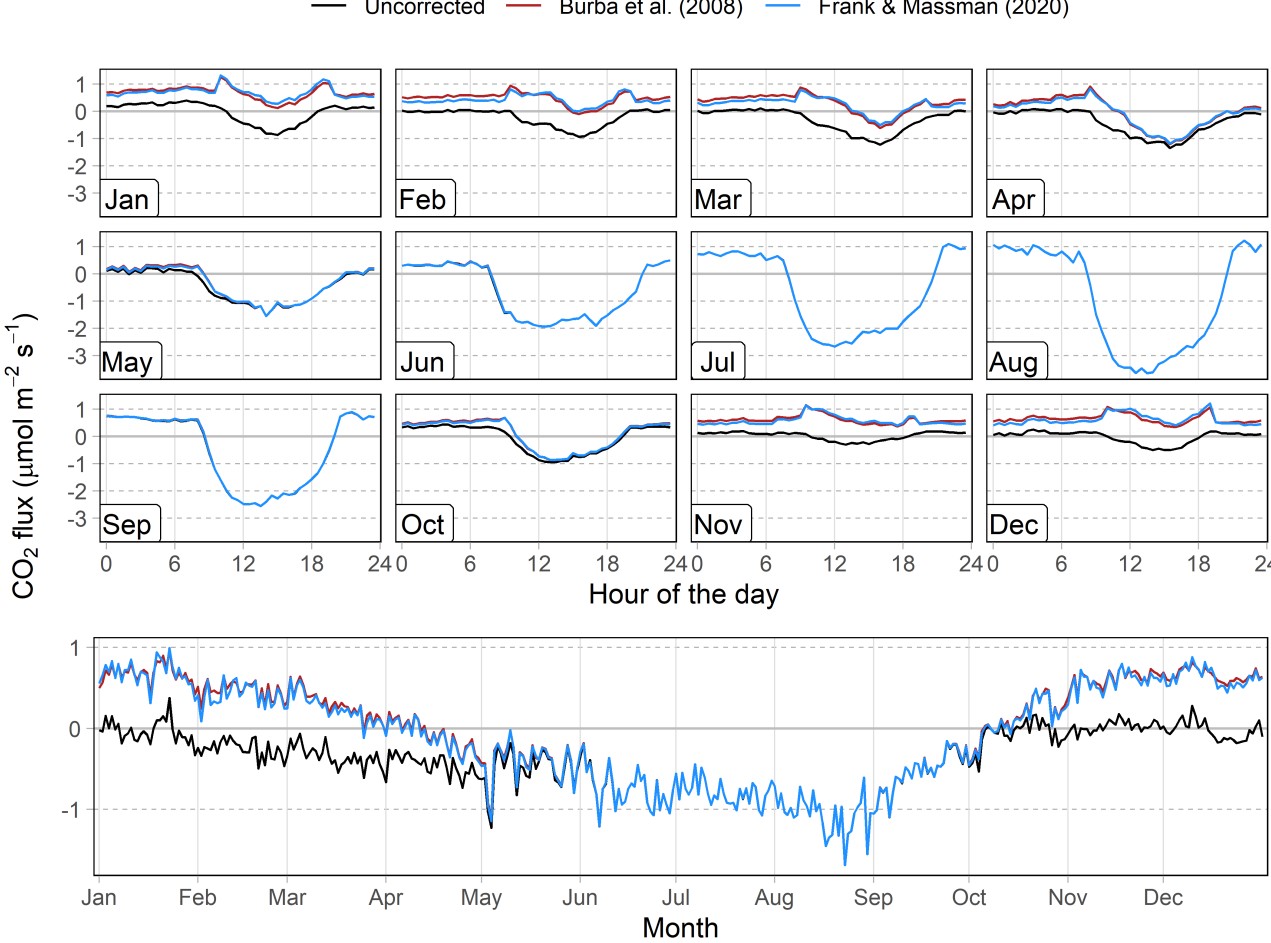

**Figure 7.** Monthly mean daily course and annual course of daily mean of the $CO_2$ flux of the years 2005 to 2019 before and after sensor self heating correction following Burba et al. (2008) and Frank and Massman (2020).

a negligible $CO_2$ uptake, so the SSH effect should become visible. Indeed, the not SSH-corrected $CO_2$ flux shows only a very small diurnal pattern with $CO_2$ uptake during daytime. This could still be a real physiological signal due to snow free patches in the EC footprint, or the SSH effect, or a combination of both. In any case, the daytime $CO_2$ uptake under these conditions and hence the SSH effect at our site is very small. In contrast, both the SSH corrections create large positive offsets in the $CO_2$ flux which are clearly an overcompensation of the SSH effect.

### 3.4 Flux uncertainty estimation

Figure 8 shows the mean diurnal and annual cycle of the $CO_2$ flux and the respective uncertainties. The two uncertainty estimates (RE, NEE_fsd) follow a distinct distribution, thereby reflecting the different sources of error they represent. As expected, the random uncertainty remains much lower than the standard deviation of the gap-filled fluxes (medians of 0.2 and 0.5 µmol m$^{-2}$ s$^{-1}$, respectively). The RE exhibits roughly the same magnitude throughout the year whereas the NEE_fsd increases with increasing flux magnitude. Concerning the diurnal course, we see lower uncertainties during nighttime and winter than during daytime and summer. The RE is generally smaller during night while during daytime, the uncertainties almost converge.

## 4  Discussion

To produce an accurate and consistent time series of $CO_2$ fluxes, we applied several correction procedures and rigorously checked for data quality constraints during the long observation period, spanning almost 15 years. Nevertheless, some uncertainties remain, mainly due to technical and logistical constraints, as well as limited documentation of the measurements.

We applied the drift correction in order to remove a systematic bias in concentration measurements. Although the correction procedure itself works well, there are certainly other effects that could reduce the effective removal of the concentration drift. The use of the Mt. Waliguan time series as input for the model, which was used to derive the offset between measured and "real" $CO_2$ concentration, may be responsible for some degree of error. First, the measurements at Mt. Waliguan represent the atmospheric background $CO_2$ concentration 1100 km NE of Nam Co, which does not necessarily mean, that they also represent the $CO_2$ concentrations at our study site. Second, the value used to estimate the offset was derived from a model. This approach somewhat smoothes the time series and generates the same annual pattern for every year while applying a constant rise in $CO_2$ concentrations. There is a good agreement between the two time series when sensor calibrations have just been conducted (red circles in 2009, 2012, 2017 and 2019 in Fig. 3). To be more precise on that, the measured daily medians remain approximately 10-15 ppm lower than the model right after user calibration was performed. An underestimation of 15 ppm around 400 ppm means about 3.75 % error in concentration, which leads to roughly 1.5 % error in flux for $CO_2$ (the % error in flux is roughly 40 % of the % error in concentration, Fratini et al., 2014). Considering that the measured concentrations are often 100 to several 100 ppm away from the (assumed) real concentration and that this causes great errors in the raw flux and the WPL correction, this correction can be assumed to greatly reduce these errors. As seen in Figs. 3 and 4, after drift correction, the mean $CO_2$ and $H_2O$ concentrations are very close to the (assumed) values. So even though not completely accurate, this strategy is expected to at least reduce the inaccuracy of the computed fluxes. Concerning the drift correction of the $H_2O$ measurements, the offset was derived from adjacent low frequency measurements of relative humidity, air temperature and air pressure. Although the approach itself is robust, there may be some degree of uncertainty due to the limited long term stability of the measurements which is claimed by the manufacturer to be "better than 1 % RH per year" (Vaisala, 2006).

During the long observation period the surrounding of the EC system was subject to rather profound changes in constructions and scientific infrastructure. The wind regime changed substantially with the construction of the two-storey main building in

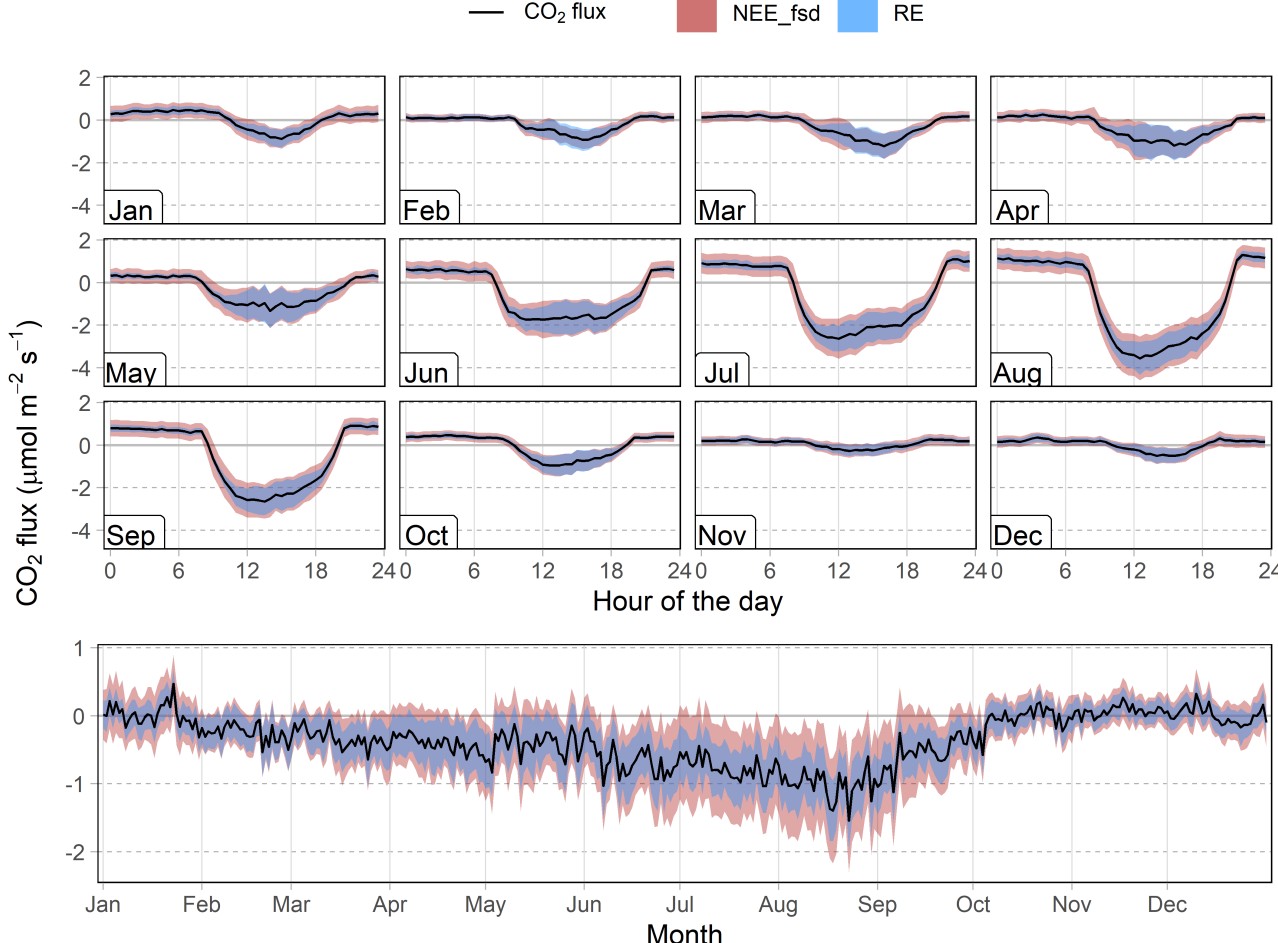

**Figure 8.** Monthly mean diurnal course and annual course of daily mean of the $CO_2$ flux and uncertainty estimates: RE is the random uncertainty following Finkelstein and Sims (2001), NEE_fsd is the standard deviation of values used for gap filling after Reichstein et al. (2005).

2010. The horizontal wind is forced to flow around this large obstacle, thereby increasing wind speeds and turbulent mixing reaching the EC system from western direction. Although the other constructions seem not to exhibit such a profound impact on the horizontal wind speed, some stronger turbulent mixing can be observed in the lee of the smaller buildings, such as the PBL container and the laboratory. To address a possible influence by human activity, fluxes which originate from the disturbed sectors should be excluded from further analyses.

The use of a pre-2010 model of the Li-7500 open path IRGA usually requires the correction for apparent off-season $CO_2$ uptake due to air density fluctuations in the measurement path of the sensor (i.e. sensor self heating correction). We found the SSH effect to be rather small at our study site and moreover, the SSH corrections following Burba et al. (2008) and Frank

and Massman (2020) clearly overcompensated the effects. Furthermore, we assume that there is a real daytime $CO_2$ uptake during winter at our study site. This could be explained by the scarce snow cover and the generally high solar radiation even during the coldest months. Measurements of surface temperature (soil temperature in 0 cm depth) show temperatures well above 0 °C during daytime in winter (mostly between 12:00 and 18:00, see Fig. A2). Plants may photosynthesize until below -3 °C, at least they do so in Antarctic tussock grass (Bate and Smith, 1983) and lichens may photosynthesize under even colder conditions (Kappen et al., 1996). In summary, we suggest that the $CO_2$ uptake during winter daytime represents a physiologically meaningful signal rather than an artifact from the SSH effect. Further research should be performed to better disentangle the two effects, hence we provide the following $CO_2$ flux time series: (1) no SSH correction applied, (2) SSH correction following Burba et al. (2008) applied, and additionally, (3) SSH correction following Frank and Massman (2020) applied.

While systematic errors can be corrected efficiently, random errors may only be quantified in order to derive the overall precision of the measurements. Figure 8 shows the monthly mean diurnal course of the $CO_2$ fluxes and their random uncertainties (RE and NEE_fsd) from every available measurement throughout the long time series (without gap-filled values). The magnitude of these errors depends strongly on the sensible heat flux (Liu et al., 2006). Hence, the random flux errors are especially pronounced during daytime and summer, when sensible heat fluxes are large. During winter (November to February), the random error oftentimes exceeds the magnitude of the fluxes. The random uncertainty estimates described above represent random flux components as well as spatial heterogeneity, temporal variability, and small meteorological variability while neglecting other sources of random flux errors such as instrument noise.

One uncertainty, about the type of alpine steppe that the fluxes are supposed to represent, still exists. The whole NAMORS station was fenced in 2006, thus preventing the otherwise ubiquitous livestock from grazing in the footprint area. Wei et al. (2012) carried out chamber measurements within and outside the fenced area at the Nam Co station during the growing seasons of 2009 and 2010. The period of 4 years of livestock exclosure significantly increased above ground biomass which possibly led to lower soil temperatures due to shading effects. While the authors did not find a significant effect on $CO_2$ emission patterns during the two growing seasons, $CO_2$ emissions tended to be less sensitive to temperature change (i.e. lower $Q_{10}$ value). Their findings are corroborated by Hafner et al. (2012) who used $^{13}C$ pulse labeling to assess the carbon cycle of a montane *Kobresia* pasture with moderate grazing and a 7-year-old grazing exclosure plot in the province of Qinghai on the north-eastern TP. While the total $CO_2$ efflux of the grazed and ungrazed grasslands remained similar, the grazing exclosure had a negative effect on organic carbon stocks in the upper 15 cm of the soil profile due to decreased total carbon input into the soil by plants and enhanced decomposition of medium and long term carbon stocks. The different processes governing the carbon cycle for grazed and ungrazed conditions have to be taken into account when drawing any conclusions on the ecosystem-atmosphere exchange from the site at Nam Co.

## 5 Related Work

Studies on carbon cycling on the central Tibetan Plateau have focused mainly on the *Kobresia* pastures (e.g, Ohtsuka et al., 2008; Babel et al., 2014; Zhang et al., 2016, 2018). The alpine *Kobresia* pastures represent an overall small sink of carbon dioxide, but its strength is highly variable within a year, as well as between several years (Gu et al., 2003; Kato et al., 2004, 2006; Saito et al., 2009). While alpine pastures have been studied extensively, the alpine steppe ecosystem has experienced relatively little interest, although it covers a much larger area. This can be explained by difficult accessibility and corresponding under-representation of (micro- ) meteorological measurements. While the principal drivers of ecosystem-atmosphere $CO_2$ exchange seem to be similar in alpine steppe and pasture ecosystems, Ganjurjav et al. (2016) showed that warming significantly stimulated plant growth in the alpine pastures, but reduced growth and diversity in the alpine steppe ecosystem. Findings for the alpine steppe ecosystem suggest overall high correlation between soil water content and $CO_2$ fluxes, while it could not be clarified, whether the alpine steppe acts as an overall sink or source of $CO_2$ (Zhu et al., 2015; Wang et al., 2016). The fluxes varied substantially between the years depending on onset and strength of the monsoonal precipitation and temperature, indicating a close correlation with the strong seasonality on the TP. Interestingly, the high solar radiation seems to hamper diurnal carbon uptake by exceeding the maximum photosynthetic capacity during noon. Wei et al. (2012) conducted chamber measurements at Nam Co, corroborating the small sink strength for the growing seasons 2009 and 2010. This study is the first to report long-term, year-round $CO_2$ fluxes from the alpine steppe ecosystem which may be used to better understand carbon cycling under accelerated climate change scenarios.

## 6 Conclusions

Here, we present the first long term eddy covariance (EC) $CO_2$ and $H_2O$ flux measurements from the alpine steppe ecosystem which covers roughly 800,000 $km^2$ on the central Tibetan Plateau. The harsh environmental conditions and the remote location at > 4500 m above sea level make continuous and high-quality measurements especially challenging. To ensure meaningful flux estimates, we applied rigorous quality filtering rules and analyzed the turbulent flow regime to identify erroneous data. We efficiently removed a drift in mean concentration measurements, possibly caused by dirt contamination in the optical path and ageing internal chemicals of the IRGA (Fratini et al., 2014). Furthermore, we found that the sensor self heating effect during cold conditions only plays a minor role at our study site. When applying the standard Burba et al. (2008) self heating correction and the revised formulations by Frank and Massman (2020), we clearly see an overcompensation of the SSH effect. High solar radiation and midday soil surface temperatures well above 0 °C suggest that the small carbon uptake during winter daytime may indeed be a physiological meaningful signal rather than an artifact. The wind direction distributions of wind speed and TKE suggest that the several buildings which were constructed in close vicinity of the tower do exert some influence on the flow regime. Fluxes originating from the disturbed areas should be excluded from further analyses as they may be compromised by human activities. Data availability of $CO_2$ fluxes after quality filtering and gap filling is quite different for individual years. While seven complete years of $CO_2$ ecosystem-atmosphere exchange are available, the filled data gaps are quite large, covering up to two months and should therefore be interpreted carefully. Unfortunately, the whole research station was fenced in 2006,

thus preventing the otherwise ubiquitous yak, goat and sheep from grazing within the footprint. While biogeochemical cycles react quite slowly on the grazing exclosure, there is certainly some influence on vegetation and soil properties which should be subject to further examination.

Nearly 15 years of consistently processed and quality controlled $CO_2$ flux data from the large but underrepresented Tibeten alpine steppe ecosystem are a valuable addition to further deepen the knowledge on carbon cycling in high alpine grassland ecosystems, which are especially vulnerable to global warming. The presented data set covers $CO_2$ and $H_2O$ fluxes with quality flags for each processing step, footprint modeling and NEE gap filling results, as well as auxiliary measurements of meteorological variables and can be accessed via https://www.doi.org/10.5281/zenodo.3733202 (Nieberding et al., 2020a) and

https://www.doi.org/10.11888/Meteoro.tpdc.270333 (Nieberding et al., 2020b). This comprehensive data set allows potential users to put the gas flux dynamics into context with ecosystem properties, potential flux drivers and allows for comparison with other data sets.

## 7   Code and data availability

The data set was uploaded to Zenodo and is freely available under https://www.doi.org/10.5281/zenodo.3733202 (Nieberding

et al., 2020a). Furthermore, the data set is available on the National Tibetan Plateau Center and can be accessed through https://www.doi.org/10.11888/Meteoro.tpdc.270333 (Nieberding et al., 2020b) after user registration on the website. The data sets are published under Creative Commons Attribution 4.0 International (CC BY 4.0) license. The R scripts used in this study are provided in the supplementary material of this manuscript.

## Appendix A:  Processing of meteorological parameters

The PBL tower in close vicinity to the EC system provides additional measurements of air temperature (Ta), relative humidity (RH), wind speed (WS) and wind direction (WD) in several heights, as well as soil temperature (Ts), soil moisture (SMC) and soil heat flux (SHF) in several depths (see Sect. 2.1). A four component radiometer provides measurements of short and long wave incoming and outgoing radiation (SWin, LWin, SWout, LWout, respectively) and a net radiometer provides global radiation (GR). Furthermore, measurements of air pressure (Pa) are available. As a first step, time periods with obviously

incorrect measurements were removed. Secondly, we set upper and lower thresholds for every measurement in order to removed physically implausible values from the time series: 30 °C < Ta < -35 °C; 100 % < RH / SMC < 0 %; 1000 W m$^{-2}$ < SHF < -500 W m$^{-2}$; 1500 W m$^{-2}$ < GR / SWin / SWout (values < 0 W m$^{-2}$ were set to zero); 410 W m$^{-2}$ < LWin < 75 W m$^{-2}$; 750 W m$^{-2}$ < LWout < 150 W m$^{-2}$; 600 hPa < Pa < 500 hPa. In order to produce a time series as complete as possible, we merged biometeorological variables when possible. The low frequency air temperature and relative humidity measurements from the

EC tower were filled step wise with the respective measurements from the different heights of the PBL tower, depending on their correlation (2 m > 4 m > 1.5 m > 10 m > 20 m). For Ts, the measurements from 0 cm depth were filled with the measurements from 10 cm and 20 cm depth. For SMC and SHF, the measurements from 10 cm depth were filled with the

respective measurements from 20 cm depth. For the short and long wave radiation components, additional measurements for the years 2016 and 2017 were available and used when needed. As a last step, data gaps up to one hour (two time-steps) were

linearly interpolated. The resulting time series were used as biometeorological input data for EddyPro and are supplied with the data set.

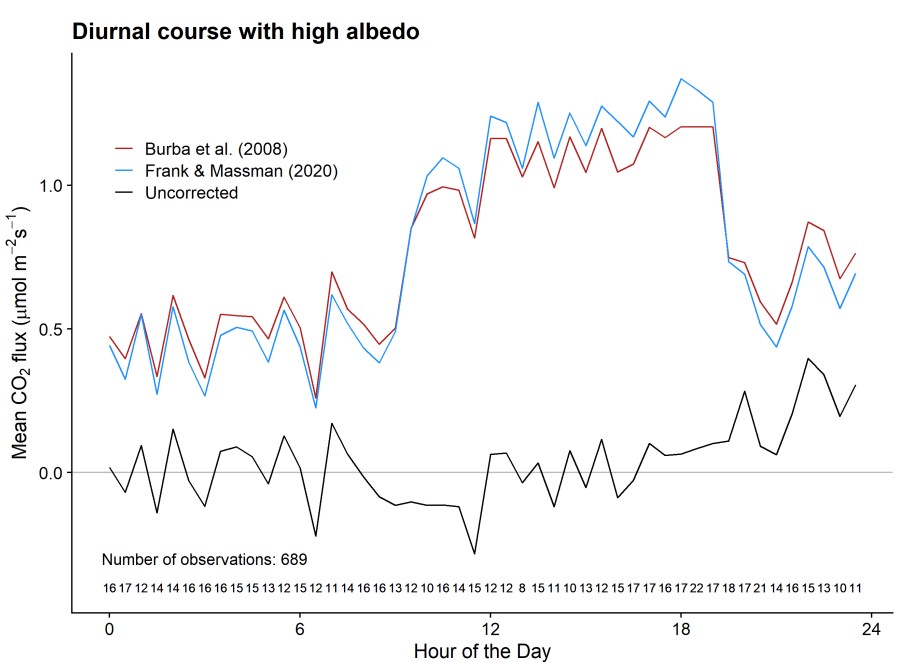

**Figure A1.** Mean diurnal course of the original and SSH corrected (Burba et al., 2008; Frank and Massman, 2020) $CO_2$ flux, during cold periods (air temperature < 0 °C) and closed snow cover (short wave albedo > 0.8). Note that a closed snow cover is rarely found at our site, therefore the number of data points is limited. Most data originate from the winter of 2006-2007.

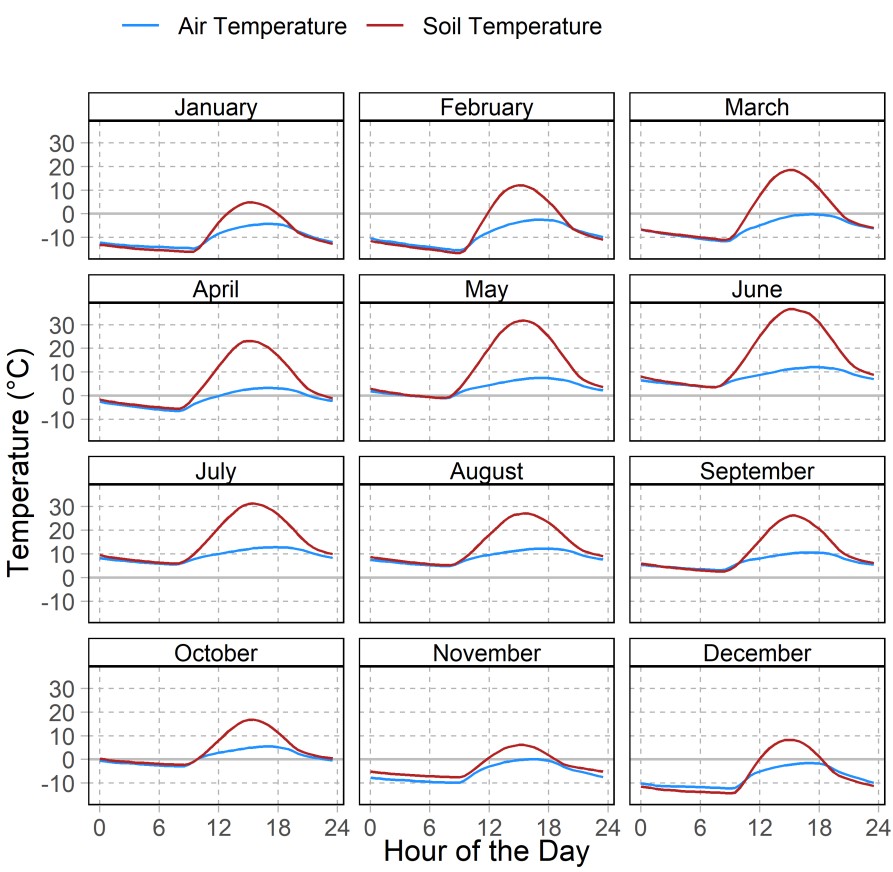

**Figure A2.** Monthly mean daily course of air and soil temperatures of the years 2005 to 2019.

**Table A1.** Data availability after individual processing and filtering steps. All units in % of the whole year, respectively

| Year | $CO_2$ fluxes | | | | | $H_2O$ fluxes | | | |
|------|-----|-----------|-------|------|-------------|------|-----------|-------|------|
|      | Raw | Wind dir. | Ustar | QC   | Gap filling | Raw  | Wind dir. | Ustar | QC   |
| 2005 | 7.3  | 6.1  | 5.0  | 4.5  | 7.4  | 7.3  | 5.8  | 4.7  | 4.2  |
| 2006 | 55.0 | 42.8 | 38.2 | 33.1 | 100  | 55.0 | 41.1 | 36.4 | 31.9 |
| 2007 | 75.6 | 61.0 | 55.0 | 48.0 | 100  | 75.5 | 61.1 | 55.0 | 48.2 |
| 2008 | 86.9 | 69.2 | 63.4 | 54.7 | 100  | 86.1 | 67.2 | 61.7 | 53.9 |
| 2009 | 92.8 | 72.0 | 65.1 | 57.0 | 100  | 92.8 | 69.7 | 62.9 | 56.1 |
| 2010 | 71.4 | 48.0 | 41.8 | 35.6 | 100  | 71.4 | 46.5 | 40.4 | 34.8 |
| · 2011 | 65.0 | 41.2 | 36.6 | 31.3 | 100 | 65.0 | 41.3 | 36.8 | 31.7 |
| 2012 | 19.2 | 11.7 | 10.4 | 9.0  | 41.3 | 19.2 | 11.6 | 10.3 | 8.9  |
| 2013 | 0.2  | 0.0  | 0.0  | 0.0  | 2.7  | 0.2  | 0.0  | 0.0  | 0.0  |
| 2014 | 26.5 | 17.8 | 16.7 | 14.1 | 32.9 | 26.2 | 17.5 | 16.4 | 13.7 |
| 2015 | 40.1 | 23.7 | 20.9 | 18.5 | 62.2 | 40.0 | 23.0 | 20.2 | 18.0 |
| 2016 | 48.2 | 30.5 | 27.8 | 23.6 | 84.9 | 48.3 | 30.0 | 27.4 | 24.0 |
| 2017 | 77.0 | 49.7 | 44.7 | 37.4 | 100  | 77.0 | 49.1 | 44.2 | 37.6 |
| 2018 | 30.5 | 15.7 | 13.7 | 11.8 | 30.8 | 30.5 | 14.8 | 13.0 | 11.3 |
| 2019 | 21.4 | 13.5 | 12.4 | 10.2 | 33.1 | 21.1 | 13.2 | 12.2 | 9.5  |

*Author contributions.* TS, CW and YM conceptualized and administered the research activity planning and execution and acquired funds for it. FN, MOA and YW conducted the investigation. FN, CW, GF and MOA analyzed the data. FN and MOA created the visualizations. FN wrote the original draft. FN, CW, GF, MOA, YW, YM and TS reviewed and edited the original draft.

*Competing interests.* The authors declare that they have no conflict of interest.

*Acknowledgements.* We thank George Burba for providing the Excel spreadsheets from which the initial calculations of the sensor self heating correction were derived. Special thanks go to Guoshuai Zhang and Binbin Wang who were a great help during field work and to Guoshuai Zhang for providing the aerial images used in Figs. 1, 2. We thank Prof. Li Jia from the Institute of Remote Sensing and Digital Earth (CAS) for providing additional radiation data. This research is a contribution to the International Research Training Group
"Geo-ecosystems in transition on the Tibetan Plateau (TransTiP)", funded by Deutsche Forschungsgemeinschaft (DFG grant 317513741 / GRK 2309). It was supported by the Second Tibetan Plateau Scientific Expedition and Research (STEP) program (2019QZKK0103), the Strategic Priority Research Program of Chinese Academy of Sciences (XDA20060101) and the National Natural Science Foundation of China (91837208). We acknowledge support by the Open Access Publication Funds of the Technische Universität Braunschweig.

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
