# Peer review of "A Long Term (2005 - 2019) Eddy Covariance Data Set of $CO_2$ and $H_2O$ Fluxes from the Tibetan Alpine Steppe"

_Earth System Science Data, 2020_

## Referee Comment (RC1) · Anonymous Referee #1 · 17 Jun 2020

I have not reviewed the data paper in the past, so admittedly can be off on some of the aspects of the review.

Dataset seems useful, well-cleaned, and extremely well documented. Usability has been helped by the detailed description of the data, explanations of their meaning, and the value of data from the unique high-elevation location of Tibetan Alpine Steppe.

In short, if a user wanted to utilize these data for their research or other assessment, they have been provided very clear information on what these data contain, how they were obtained, quality controlled, and re-processed.

---

## Referee Comment (RC2) · Anonymous Referee #2 · 1 Jul 2020

The manuscript presents a 15 year time series from eddy covariance and meteorological data from the Tibetan Alpine Steppe. Due to the remote site and harsh environmental conditions the timeseries in unfortunately full with gaps and many needed maintenances have not been done. Additionally, the fetch was strongly disturbed by construction of buildings. Still the authors challenged these hard data conditions and tried to extract all that was possible out of these data. First of all, I want to express my deeps respect for the thorough work the co-authors did to create this data set. Obviously, a lot of brains were put together to solve the many problems present in the original data set and time series. I carefully read the manuscript, downloaded the data files and R-codes. It is a great effort in terms of reproducibility of the data which I

very much appreciate. I want to highlight that I truly think these data are worth to being published. My impression is that maybe too much data massaging (with the best intentions I'm sure) was done so it looks like more than it is. The manuscript is well written and clearly structured, the data are well documented and the provided code is in a good shape. Figures n the manuscript nicely illustrated. Due to the points below, I recommend a major revision. I'm want to see the manuscript again and I'm very much looking forward to see your comments and suggestions. I'm sure you can tackle all the points and solve them.

Please find my comments in the attached pdf.

Please also note the supplement to this comment:
https://essd.copernicus.org/preprints/essd-2020-63/essd-2020-63-RC2-supplement.pdf

**Supplement:**

The manuscript presents a 15 year time series from eddy covariance and meteorological data from the Tibetan Alpine Steppe. Due to the remote site and harsh environmental conditions the timeseries in unfortunately full with gaps and many needed maintenances have not been done. Additionally, the fetch was strongly disturbed by construction of buildings. Still the authors challenged these hard data conditions and tried to extract all that was possible out of these data.

First of all, I want to express my deeps respect for the thorough work the co-authors did to create this data set. Obviously, a lot of brains were put together to solve the many problems present in the original data set and time series.

I carefully read the manuscript, downloaded the data files and R-codes. It is a great effort in terms of reproducibility of the data which I very much appreciate. I want to highlight that I truly think these data are worth to being published. My impression is that maybe too much data massaging (with the best intentions I'm sure) was done so it looks like more than it is.

The manuscript is well written and clearly structured, the data are well documented and the provided code is in a good shape. Figures n the manuscript nicely illustrated.

Due to the points below, I recommend a major revision. I'm want to see the manuscript again and I'm very much looking forward to see your comments and suggestions. I'm sure you can tackle all the points and solve them.

Major points:

WPL and SSH correction:

After WPL and SSH you still have a diurnal cycle in the $CO_2$ data even during winter. Obviously, this pattern is not real but this is not at all discussed in the manuscript. This is most likely a WPL correction effect and not a physiological meaningful signal.

The buildings:

The wind disturbance due to the buildings is basically argued away even though the problem still remains. The easiest solution would be the removal of the wind direction $230 - 300$ degree. Maybe account for the years and increase the angles based on the years when the respective buildings were constructed. I fully understand that you want to keep as many data as possible but the undisturbed wind field is not given at all if there are massive buildings so close to the tower. Additionally, the footprint calculation have basically no value as the assumptions of a homogeneous terrain and wind flow are strongly violated. It is a shame that the buildings were build there. The consequence is that you can't use the data and that must be faced. Further, I assume the buildings are creating heat and $CO_2$, greenhouse maybe even contribute to a $CO_2$ sink. All these influences can't be accounted for that is why it is required to remove these data.

This simple plot gives some nice indication when things changed and how they influenced the wind field of the sonic. A slightly tilted sonic in flat terrain would have a sine shape. Here you can see the obstacles and how they influence the wind field and when things changed. This could let you also think about the size of the sectors for the planar fit methods (just as an idea).

[Figure]

The wind coming from the back of the sonic in a set up as you have (CSAT) it should generally be removed due to flow distortion. That would be something like 350 – 10 degree.

CO2 concentration correction:

Here a novel data correction method is introduced. The correction seems reasonable but it has not been tested, nor have uncertainties and problems been investigated? There are quite some differences between Mauna Loa and e.g. Mt. Waliguan the closest station to NAMORS I saw at https://www.esrl.noaa.gov/gmd/. What would be the differences using one or the other for the flux calculations? Of course, one can run the analysis for 50 other atmospheric CO2 background stations and see how the fluxes change but we are still missing the truth at the site. If such a method is to be used it must be thoroughly evaluated and this has not happened. Even if this correction is valid for the flux calculation it is for sure not valid to sell the resulting CO2 concentrations as the measured concentrations. If the data have not been measured by the instrument the qc-flag must be 2 and not 0. I would also like to address one point which I guess Mr. Fratini can help with or at least validate. His paper from 2014 was done with a LI7200 (and a LI7000 as reference) which is an enclosed instrument using an inlet and in best case a filter that ensures that the inside of the sensor stays clean. As you described it you used a LI7500 open path sensor that besides the changes in the offset and the span is also highly affected by the dirt accumulating on the windows. But this effect cannot be simply calculated back, correct? If I remember correctly the LI7500 puts out the "automatic gain control" (AGC) as an indication how clean/dirty the windows are. And, there are recommendations to which AGC-value data should be used or discarded. If you have any change to get this value out of the raw data t would for sure help you to better QC the data.

The CO2 concentration data in the data file are now following on average Mauna Loa but can we assume this is correct? The half hourly data show a gigantic scatter in mixing ratios between 0 and 600 ppm. Throughout the measuring period there are values of 0 in CO2 concentration. This is interesting because when using the "qc_co2_flux_composite" filter and only select data when "qc_co2_flux_composite" is equal to zero there are many of these 0-concentration data left. In fact, there are 612 data point for which CO2 concentrations are below 300 ppm (including zero-values) or above 600ppm and fluxes seem to be of high quality. This means that the fluxes have been calculated from an average concentration of 0. Does that make any sense? I would say no. You might

say who cares about 612 points in a data set of 241178 but it shows that the QC scheme is still including errors.

I'm honestly not convinced by this correction method especially because it was not developed for an instrument where changes in absorption might also arise from dirt on the windows. And because it was not tested and cannot be evaluated with the current data set. I'm sorry for being so negative about this correction but I hope I made my point clear and you share my point of view.

$H_2O$ concentration:

The $H_2O$ concentrations provided in the data-file are not the once from the LI7500 but the once from the biomet data, i.e. the temperature and relative humidity sensor. This might be okay for a normal data set where no issues are present with drifts, dirty windows, concentration etc. But here I would highly recommend to provide or at least look at the water vapor concentration of the LI7500. When you use EddyPro for processing I think the only way to get the true LI7500 $H_2O$ concentrations is when you run the processing without providing the biomet file. The point here is that you can't use the concentration as a quality criteria. You can actually see that by looking at the number of qc_co2_mixing_ratio_composite and qc_h20_mixing_ratio_composite. The number of bad data for qc_co2_mixing_ratio_composite (==2) is 12730 and for qc_h2o_mixing_ratio_composite (==2) is 203. If the concentration of the one is bad usually also the other one is bad. Especially when this is due to dirty windows, precipitation, snow frost, etc… I really encourage you to use the real LI7500 water vapor concentrations to select a criterion to remove bad data and also bad fluxes of h2o.

In principle I would recommend to provide the raw $CO_2$ and $H_2O$ concentrations and the corrected once.

Minor comments

Please include the countries to which the southern and western part of the TP belongs. I guess Nepal, Pakistan, India and Bhutan.

The sine-cosine model is not explained. Why not directly using a spline function or even a moving average? Did you use the flask samples or the continuous? The pattern of the $CO_2$ concentrations is not really a sine or cosine.

I think the u* filtering should be applied. Just because there s wind does not mean there is no relationship. For grassland values around u* values of 0.1 m/s are not so uncommon and that accounts in your data set for 15% of all u* values. The red line in the plot shows the cumulated density function multiplied by 3000 to fit the scale. Green vertical line is at u*== 0.1 m/s.

Formular 7 and 8 you use mu and sigma which are usually the population mean and its standard deviation. I know you took it from the paper of Burba but there are plenty of other variables one can use.

For the uncertainty analysis of the WPL I have only a gut feeling that this is wrong but it would be good if you would get some statisticians input and explain why this is valid to do. In principle each value n the formula has an uncertainty e.g. Ta which propagates in Cp and rho. Sorry for not being more helpful on this one.

But generally, for the overall uncertainty I would rather take the NEE_fsd to calculate the uncertainty of the fluxes. It is including not only the random error but also temporal variability and

spatial heterogeneity. There is a paper comparing these uncertainties with each other I think in a savanna (sorry I can't recall the author maybe worth a look).

[Figure]

The sentence in line 425 "The wind direction distributions of wind speed and TKE, as well as the analysis of cumulative footprints suggest that the several buildings which were constructed in close vicinity of the tower do exert some influence on the flow regime while not violating basic EC assumptions. Nevertheless, fluxes originating mainly from the disturbed areas should be excluded from further analyses as they may be compromised by human activities."

You have all indications that the flow was clearly disturbed and you still conclude that the assumptions of eddy covariance are met? How does that go together?

For the data description file some more info on the uncertainties would be great. The differences are not directly clear to the reader.

I'm happy to follow up on this and to keep discussing

Find below my code for reading the data and making some plots. Nothing fancy but only fair to provide.

**check data for essd manuscript**

library(lubridate)

```
df_fromEPO<-function(FPath){

 df=read.csv(paste(FPath,sep =""), header=FALSE,skip=1,sep="\t",blank.lines.skip=FALSE,
fill=FALSE,strip.white=TRUE);

 cnames <- names(read.csv(FPath, nrows=1, skip=0, sep="\t"))

 names(df)<-cnames

 #convert date format with lubridate package

 df$date=as.character(df$date);

 df$time=as.character(df$time);

 as.character(df$rDate);

 df$rDateChr=paste(df$date,df$time);

 df$rDate=parse_date_time(df$rDateChr,"%y%m%d%H%M",tz="GMT");

 #delete missing values with error code -9999

 df[df==-9999]=NA

 return(df1)
}

FPath="D:/Reviews/2020ESSDLongTermECDataCO2H2OTibetanSteppe/NAMORS_EC_2005-2019.txt"

df=df_fromEPO(FPath)
plot(df$rDate,df$co2_flux,pch='.')
plot(df$rDate[df$NEE_fqc==0],df$NEE_f[df$NEE_fqc==0],pch='.')
points(df$rDate,df$co2_flux,pch='.',col='red')
plot(df$rDate,df$co2_mixing_ratio,pch='.')
plot(df$rDate,df$co2_mixing_ratio,pch='.',ylim=c(0,500))

IDX=which(df$rDate>="2017-05-01" & df$rDate<="2018-01-01")
plot(df$rDate[IDX],df$co2_mixing_ratio[IDX],pch='.',ylim=c(0,600))
```

```r
IDX=which(df$rDate>="2017-08-31" & df$rDate<="2017-09-03")

plot(df$rDate[IDX],df$co2_mixing_ratio[IDX],pch=20,ylim=c(0,600))

plot(df$rDate[IDX],df$Ta[IDX],pch=20,ylim=c(273,290))

IDXQC=which(df$qc_co2_mixing_ratio_composite==0)

plot(df$rDate[IDXQC],df$co2_mixing_ratio[IDXQC],pch='.',ylim=c(0,600))

IDXQC=which(df$qc_co2_flux_composite==0)

plot(df$rDate[IDXQC],df$co2_mixing_ratio[IDXQC],pch='.',ylim=c(0,600))

length(which(df$qc_co2_flux_composite==0 & (df$co2_mixing_ratio<=300 |
df$co2_mixing_ratio>=600)))

plot(df$rDate,df$h2o_mixing_ratio,pch='.',ylim=c(0,20))

plot(df$wind_dir,df$w_unrot,pch='.')

abline(h=0,col='red')

plot(df$wind_dir,df$w_unrot/df$wind_speed,pch='.')

**check flow angle based on wind direction and year**

df$YYYY=year(df$rDate)

DateUni=unique(df$YYYY)

RainCols=rainbow(n=length(DateUni))

for (i in DateUni){

 print(i)

 if (i == 2005){
```

```
plot(df$wind_dir[df$YYYY==i],(df$w_unrot[df$YYYY==i]/df$wind_speed[df$YYYY==i]),pch='.',col=Rai
nCols[i-2004],ylim=c(-0.2,0.2),ylab='w_unrot/wind_speed',xlab='wind direction')

  }else{

points(df$wind_dir[df$YYYY==i],(df$w_unrot[df$YYYY==i]/df$wind_speed[df$YYYY==i]),pch='.',col=R
ainCols[i-2004])

  }

}

legend('top',bty='n',xpd=NA,inset=c(0.5,-0.14),horiz=F,ncol =
8,pch=20,col=RainCols,legend=DateUni)

abline(h=0,lwd=2)

SinVals=seq(0,2*pi,length.out = 361)

xvals=seq(0,360,1)

points(xvals,sin(SinVals)*-0.1,type='l',lwd=2)

**check u-star distributon**

plot(df$rDate,df$ustar,pch='.')

hist(x = df$ustar,breaks = seq(0,5,0.01),xlim=c(0,1.5))

uStarEcdf=ecdf(x = df$ustar)

points(x=seq(0,1.5,0.01),uStarEcdf(v = seq(0,1.5,0.01))*3000,type='l',col='red',lwd=2)

abline(v=0.1,col='green',lwd=2)

uStarEcdf(0.1)*100 # percent of u*-values <= 0.1 m/s

plot(df$rDate,df$qc_co2_mixing_ratio_composite,pch='.')

points(df$rDate,df$qc_h2o_mixing_ratio_composite,pch='.',col='red')

length(which(df$qc_h2o_mixing_ratio_composite==2))

length(which(df$qc_co2_mixing_ratio_composite==2))
```

---

## Author Response (AR2)

We highly appreciated the critical but extremely constructive reviewer comments and their thoughtful suggestions. Based on these comments we carefully revised our manuscript. Below you will find our point-by-point response to the reviewer´s comments and suggestions, as well as the corresponding changes in the manuscript. The revised data set was uploaded to the repositories and is available under https://doi.org/10.5281/zenodo.3733202 and https://doi.org/10.11888/Meteoro.tpdc.270333 (Version 2).

In the name of all co-authors,

Felix Nieberding

**Response to Anonymous Referee #1**

Dear referee, thank you very much for your positive assessment of our manuscript. It is appreciated that the added value of our study is seen by the scientific community, especially with regard to the extensive documentation of the data set.

**Response to Anonymous Referee #2**

**Major points:**

1. **WPL and SSH correction:**

**Reviewer comment 1:** After WPL and SSH you still have a diurnal cycle in the CO2 data even during winter. Obviously, this pattern is not real but this is not at all discussed in the manuscript. This is most likely a WPL correction effect and not a physiological meaningful signal.

**Response to reviewer comment 1:** We thoroughly re-considered the SSH correction. In the process we found that the correction approach by Oechel et al. (2014) which we used before should not be applied to our data because not all requirements are fulfilled at our study site. Hence it was removed from the data set and the manuscript. The de facto standard SSH correction following Burba et al. (2008) has been used widely at cold ecosystem flux sites (e.g. Miller et al., 2011; Webb et al., 2016) but has also been modified frequently because it has been shown to produce unsatisfactory results (Kittler et al., 2017; Oechel et al., 2014). In order to overcome this problem, we now provide the following CO2 flux time series: (1) no SSH correction applied, (2) SSH correction following Burba et al. (2008) applied, and additionally, (3) SSH correction following Frank and Massman (2020) applied. The latter approach was chosen because it corrects significant errors of the method of Burba et al. (2008).

[Figure]

*Fig. 1: The new Fig. 7 in the manuscript: Monthly mean daily course and annual course of daily mean of the CO2 flux of the years 2005 to 2019 before and after sensor self heating correction following Burba et al. (2008) and Frank and Massman (2020).*

The monthly mean diurnal course of the three CO2 flux time series clearly show the effect of the SSH corrections during the cold months (Fig. 1). The effect of the two approaches of Burba et al. (2008) and Frank and Massman (2020) are very similar. We see various problems associated with the SSH corrections: (1) the corrections create strong artifacts during the transition between day and night, (2) SSH-corrected night-time CO2 fluxes during the cold months are very high – at about the same level as night-time CO2 fluxes during summer – suggesting an over-correction of SSH effects, and (3) the winter-time diurnal course of CO2 flux with day-time uptake of CO2 – which is assumed to be the effect of the SSH – does not disappear, but the daytime CO2 flux is merely offset by what seems to be a more or less constant flux value. This leads us to the conclusion that the effect of the SSH is very small at our site and that the application of the standard correction (Burba et al., 2008) and its improved version (Frank and Massman, 2020) lead to an undue over-correction of this effect. Furthermore, it leads us to the assumption that there is a real day-time CO2 uptake during winter. This could be explained by the scarce snow cover and the generally high solar radiation at our site. Measurements of surface temperature (soil temperature in 0 cm depth) show temperatures well above 0 °C during daytime in winter (mostly between 12:00 and 18:00, see Fig. 2). Plants may photosynthesize until below -3 °C, at least they do so in Antarctic tussock grass (Bate and Smith, 1983). Lichens may photosynthesize under even colder conditions (Kappen et al., 1996).

[Figure]

*Fig. 2: Monthly mean diurnal curse of air and soil surface temperature.*

To test our conclusion about the SSH effect at our site, we calculated the mean diurnal course of $CO_2$ fluxes during cold periods with a closed snow cover (Fig. 3). Under these (rare) conditions, we expect a negligible $CO_2$ uptake, so the SSH effect should become visible. Indeed, the not SSH-corrected $CO_2$ flux shows only a very small diurnal pattern with $CO_2$ uptake during daytime. This could still be a real physiological signal due to snow free patches in the EC footprint, or the SSH effect, or a combination of both. In any case, the daytime $CO_2$ uptake under these conditions and hence the SSH effect at our site is very small. In contrast, both the SSH corrections create large positive offsets in the $CO_2$ flux which are clearly an overcompensation of the SSH effect. In summary, we suggest that the $CO_2$ uptake during winter represents a physiologically meaningful signal and not an artifact from incomplete SSH corrections. Nevertheless, $CO_2$ fluxes with applied SSH correction will be part of the updated data set. A paragraph describing the issue was introduced into Sect. 2.5, 3.3 and this topic was raised in the discussion and conclusions as well.

[Figure]

*Fig. 3: Mean diurnal course of the original and SSH corrected (Burba et al., 2008; Frank and Massman, 2020) CO2 flux, during cold periods (air temperature < 0 °C) and closed snow cover (short wave albedo > 0.8). Note that a closed snow cover is rarely found at our site, therefore the number of data points is limited. Most data originate from the winter of 2006-2007.*

**Changes in the manuscript concerning comment 1:** The SSH correction following Oechel et al. (2014) was discarded throughout the manuscript.

We added a description of the revised formulation following Frank and Massman (2020) to the Methods Sect. 2.5 *Sensor self heating correction*: "In a recent publication, Frank and Massman (2020) tested the correction procedure for a "cold, windy, high-elevation mountainous site" and found inconsistencies in the Burba et al. (2008) correction: (1) The Burba et al. (2008) correction contains boundary layer adjustment terms for non-flat surfaces but the top- and bottom surfaces of the Li-7500 are flat. This leads to an underestimation of the surface heat fluxes which is an order of magnitude too small. (2) The weightings of the bottom and top surface heat fluxes are "improbable and an order of magnitude too large" (Frank and Massman, 2020). While these errors canceled out during the study of Frank and Massman (2020), this may not be the case for other field sites. Following the recommendations of Frank and Massman (2020), we discarded the boundary layer adjustment terms for non-flat surfaces from the calculations and applied their newly calculated weightings of the bottom and top surface heat fluxes, thus emphasizing the role of the spar in self heating. We first reproduced the Burba et al. (2008) correction like it is implemented in EddyPro and then adjusted the equations as described above."

Correspondingly, we redesigned Fig. 7 in the manuscript (Fig. 1 above) and the Results Sect. 3.3 *Sensor self heating correction* was completely reformulated: "The monthly mean diurnal course of the three CO2 flux time series in Fig. 7 clearly shows the effect of the sensor self heating correction during cold conditions (air temperature < 0 °C). The effect of the correction procedure following Burba et al. (2008) and the revised equations of Frank and Massman (2020) are very similar. We see various problems

associated with the SSH corrections: (1) the corrections create strong artifacts during the transition between day and night, (2) SSH-corrected nighttime $CO_2$ fluxes during the cold months are very high – at about the same level as the nighttime $CO_2$ fluxes during summer – suggesting an over-correction of SSH effects, and (3) the winter-time diurnal course of $CO_2$ flux with daytime uptake of $CO_2$ – which is assumed to be the effect of the SSH – does not disappear, but the daytime $CO_2$ flux is merely offset by what seems to be a more or less constant flux value. This leads us to the conclusion that the effect of the SSH is very small at our site and that the application of the standard correction (Burba et al., 2008) and its improved version Frank and Massman (2020) lead to an undue over-correction of this effect. To test our conclusion about the SSH effect at our site, we calculated the mean diurnal course of $CO_2$ fluxes during cold periods with a closed snow cover (Fig. A1). Under these (rare) conditions, we expect a negligible $CO_2$ uptake, so the SSH effect should become visible. Indeed, the not SSH-corrected $CO_2$ flux shows only a very small diurnal pattern with $CO_2$ uptake during daytime. This could still be a real physiological signal due to snow free patches in the EC footprint, or the SSH effect, or a combination of both. In any case, the daytime $CO_2$ uptake under these conditions and hence the SSH effect at our site is very small. In contrast, both the SSH corrections create large positive offsets in the $CO_2$ flux which are clearly an overcompensation of the SSH effect."

And this issue was raised in the Discussion (Sect. 4): "We found the SSH effect to be rather small at our study site and moreover, the SSH corrections following Burba et al. (2008) and Frank and Massman (2020) clearly overcompensated the effects. Furthermore, we assume that there is a real daytime $CO_2$ uptake during winter at our study site. This could be explained by the scarce snow cover and the generally high solar radiation even during the coldest months. Measurements of surface temperature (soil temperature in 0 cm depth) show temperatures well above 0 °C during daytime in winter (mostly between 12:00 and 18:00, see Fig. A2). Plants may photosynthesize until below -3 °C, at least they do so in Antarctic tussock grass (Bate and Smith, 1983) and lichens may photosynthesize under even colder conditions (Kappen et al., 1996). In summary, we suggest that the $CO_2$ uptake during winter daytime represents a physiologically meaningful signal rather than an artifact from the SSH effect. Further research should be performed to better disentangle the two effects, hence we provide the following $CO_2$ flux time series: (1) no SSH correction applied, (2) SSH correction following Burba et al. (2008) applied, and additionally, (3) SSH correction following Frank and Massman (2020) applied."

And in the Conclusions (Sect. 5): "Furthermore, we found that the sensor self heating effect during cold conditions only plays a minor role at our study site. When applying the standard Burba et al. (2008) self heating correction and the revised formulations by Frank and Massman (2020), we clearly see an overcompensation of the SSH effect. High solar radiation and midday soil surface temperatures well above 0 °C suggest that the small carbon uptake during winter daytime may indeed be a physiological meaningful signal rather than an artifact."

The Figs. 2 and 3 were added to the manuscript where they correspond to Figs. A1 and A2, respectively.

**2. The buildings:**

**Reviewer comment 2:** The wind disturbance due to the buildings is basically argued away even though the problem still remains. The easiest solution would be the removal of the wind direction 230 – 300 degree. Maybe account for the years and increase the angles based on the years when the respective buildings were constructed. I fully understand that you want to keep as many data as possible but the undisturbed wind field is not given at all if there are massive buildings so close to the tower. Additionally, the footprint calculation have basically no value as the assumptions of a homogeneous terrain and wind flow are strongly violated. It is a shame that the buildings were build there. The consequence is that you can't use the data and that must be faced. Further, I assume the buildings are creating heat and CO2, greenhouse maybe even contribute to a CO2 sink. All these influences can't be accounted for that is why it is required to remove these data. This simple plot gives some nice indication when things changed and how they influenced the wind field of the sonic. A slightly tilted sonic in flat terrain would have a sine shape. Here you can see the obstacles and how they influence the wind field and when things changed. This could let you also think about the size of the sectors for the planar fit methods (just as an idea). The wind coming from the back of the sonic in a set up as you have (CSAT) it should generally be removed due to flow distortion. That would be something like 350 – 10 degree.

**Response to reviewer comment 2:** The sine shape of the wind direction vs. unrotated vertical wind component indeed hints at a vertical tilt of the sonic anemometer that occurred during 2006 and 2007. Furthermore, the orientation of the sonic anemometer was changed from 135 degrees to 200 degrees in 2009. This was accounted for by calculating the planar fit coordinate rotation only for times when the orientation of the sonic anemometer remained constant. The dates were derived visually by analyzing the second rotation angle (pitch), estimated from a preliminary raw data processing using the double rotation method. The planar fit sectors were chosen to account for possibly disturbed turbulence from the direction of the buildings and the sensor attachment to the mast. In the paper we state that the planar fit sectors are 0° - 80°, 80° - 230 ° and 230° - 360 ° but in fact we used the following sectors: 80° - 240°, 240° - 320° and 320° - 80°. We sincerely apologize for this error and corrected it in the manuscript.

As the reviewer pointed out correctly, the assumption of a homogeneous terrain and undisturbed flow regime may not be fulfilled for all sectors. Hence, we agree that it is difficult to exclude data based on a footprint model that relies on these assumptions. We agree with the reviewer that the data from the disturbed sectors has to be excluded. We introduced an additional quality flag (qc_wind_dir), indicating whether a 30-min flux originates from a disturbed wind sector. We assume the turbulent signal to be disturbed due to (i) the back of the Csat3, (ii) the PBL container and (iii) the main buildings. The correcponding times and wind directions can be found in Tab. 1. These fluxes are excluded from subsequent analyses.

*Table 1 Disturbed wind sectors*

| From | To | Back of USA | PBL container | Main buildings |
|---|---|---|---|---|
| 2005-12-04 | 2009-06-30 | 305°–325° | - | 260°–280° |
| 2009-06-30 | 2010-01-30 | 10°–30 ° | 30°–50° | 260°–280° |
| 2010-01-30 | 2011-12-31 | 10°–30 ° | 30°–50° | 250°–300° |
| 2012-01-01 | 2018-12-31 | 10°–30 ° | 30°–50° | 250°–315° |

| 2019-01-01 | 2019-09-07 | 10°–30 ° | 30°–50° | 245°–315° |

**Changes in the manuscript concerning comment 2:** The footprint calculation was completely omitted throughout the manuscript.

In the Sect 2.1 *Site description and measurements* we added information about the change of the anemometer orientation and the installation of the KH50: "In June 2009, the sonic anemometer alignment was changed from 135 degrees to 200 degrees. In 2010, a KH50 krypton hygrometer was installed but the data is not available due to quality constraints."

In the Sect. 2.2 *Raw data processing* we added information about the vertical tilt of the USA: "During 2006 and 2007 the sonic anemometer exhibited a step wise downward vertical tilt of up to 13 degrees. This was accounted for by calculating the planar fit coordinate rotation only for times when the orientation of the sonic anemometer remained constant. The dates were derived visually by analyzing the second rotation angle (pitch), estimated from a preliminary raw data processing using the double rotation method."

The Methods Sect. 2.5 *Wind field analysis* was omitted and the information about the disturbed sectors was added to Sect. 2.4 *Quality filtering*: "During the long measuring period, spanning nearly 15 years, several buildings and scientific infrastructure were constructed in close vicinity of the eddy covariance tower. During the development of the NAMORS, from the foundation with only a few tents in 2005 to a well-equipped research station in 2019, we approximated five times with significant changes in constructions. In 2009 the PBL container, the shed and the solar panel were set up. In 2010 the main building and the green house were constructed. In 2012 the shed was rotated to become the laboratory and the tool shed next to the greenhouse was added. Finally, in 2019 the garage was relocated and extended south of the laboratory and the solar panels were removed. To assess possible influences on the flow and turbulence regime, we analyzed the wind direction distribution of the mean wind speed and the turbulent kinetic energy. We accounted for possibly disturbed turbulence, by applying the planar fit axis rotation for three different wind sectors during flux calculation (see Sect. 2.2). Furthermore, we generated a quality flag (qc_wind_dir) indicating whether a flux originates from a disturbed sector. Table 2 shows the disturbed sectors which were excluded from subsequent calculations."

The Results Sect. 3.3 *Wind field analysis* was omitted and partially merged with Sect. 3.2 *Data availability and quality filtering.*

**3.  CO2 concentration correction:**

**Reviewer comment 3:** Here a novel data correction method is introduced. The correction seems reasonable but it has not been tested, nor have uncertainties and problems been investigated? There are quite some differences between Mauna Loa and e.g. Mt. Waliguan the closest station to NAMORS I saw at https://www.esrl.noaa.gov/gmd/. What would be the differences using one or the other for the flux calculations? Of course, one can run the analysis for 50 other atmospheric CO2 background stations and see how the fluxes change but we are still missing the truth at the site. If such a method is to be used it must be thoroughly evaluated and this has not happened. Even if this correction is valid for the flux calculation it is for sure not valid to sell the resulting CO2 concentrations as the measured concentrations. If the data have not been measured by the instrument the qc-flag must be 2 and not 0. I would also like to address one point which I guess Mr. Fratini can help with or at least validate. His paper from 2014 was done with a LI7200 (and a LI7000 as reference) which is an enclosed instrument using an inlet and in best case a filter that ensures that the inside of the sensor stays clean. . As you described it you used a LI7500 open path sensor that besides the changes in the offset and the span is also highly affected by the dirt accumulating on the windows. But this effect cannot be simply calculated back, correct?  If I remember correctly the LI7500 puts out the "automatic gain control" (AGC) as an indication how clean/dirty the windows are. And, there are recommendations to which AGC-value data should be used or discarded. If you have any change to get this value out of the raw data t would for sure help you to better QC the data. The CO2 concentration data in the data file are now following on average Mauna Loa but can we assume this is correct? The half hourly data show a gigantic scatter in mixing ratios between 0 and 600 ppm. Throughout the measuring period there are values of 0 in CO2 concentration. This is interesting because when using the "qc_co2_flux_composite" filter and only select data when "qc_co2_flux_composite" is equal to zero there are many of these 0-concentration data left. In fact, there are 612 data point for which CO2 concentrations are below 300 ppm (including zero-values) or above 600ppm and fluxes seem to be of high quality. This means that the fluxes have been calculated from an average concentration of 0. Does that make any sense? I would say no. You might say who cares about 612 points in a data set of 241178 but it shows that the QC scheme is still including errors. I'm honestly not convinced by this correction method especially because it was not developed for an instrument where changes in absorption might also arise from dirt on the windows. And because it was not tested and cannot be evaluated with the current data set. I'm sorry for being so negative about this correction but I hope I made my point clear and you share my point of view.

**Response to reviewer comment 3:** We thank the reviewer for raising these critical questions. However, we only partially agree with the reviewer´s opinion. In the following, we will address the reviewer´s comments separately and try to explain, why the correction is still valid and even more, enhances the overall accuracy (and precision) of the presented data set.

**3.1  Applicability of the drift correction method**

The drift correction procedure that was applied to our data set is not a novel approach. Fratini et al. (2014) introduced, described and tested the correction and it has been implemented in the Integrated Carbon Observation System (ICOS) raw data processing protocol (Sabbatini et al., 2018). It is derived analytically from the technical characteristics of the LiCor IRGAs with dual wavelength – single path design (Li-7500 and Li-7200), and it has a very clear rationale. In fact, the correction is nothing else than an analytically rigorous re-calibration of the raw data. The correction can be applied regardless of the reason for the concentration drift, be it aging of analyzer components, thermal effects, or – as

in our case – contamination of the windows in the optical path of the sensor. The concentration drift results from an absorptance offset that causes a shift of the analyzer operating point to a different region on the absorptance-concentration calibration curve. Because this calibration curve is nonlinear, a change of the operating point leads not only to the observed offset of the mean measured gas density from the real value but also to a bias in the evaluation of density fluctuations, i.e. a change of the "span calibration" (see Fig. 2 in Fratini et al., 2014). The drift correction addresses both these effects and eliminates the associated biases. In fact, the correction procedure is basically valid for any instrument with curvilinear calibration for which a reference concentration can be established.

**3.2   CO2 concentration reference time series**

We agree that the drift correction depends on the availability of a CO2 reference, i.e. un-biased, gas concentration time series. While this is easy to achieve for H2O, where the reference can be calculated from a meteorological air temperature and relative humidity probe, it is difficult to achieve for CO2 at our site. In our approach, we used the concentration time series from the Mauna Loa observatory because it provided the best temporal coverage and resolution for the time period of our data set. In order to generate the reference CO2 time series at 30-minute resolution we fitted the following model to the Mauna Loa concentration data

$CO_{2\ ref} = p_1 + p_2 * t + p_3 * \cos(2*\pi*t/365) + p_4 * \sin(2*\pi*t/365) + p_5 * \cos(4*\pi*t/365) + p_6 * \sin(4*\pi*t/365),$

where t is the decimal time in days and $p_i$ are the fit parameters. The rationale behind the use of this model rather than a linear or spline interpolation was (1) to mimic the general pattern of the atmospheric background CO2 concentration while excluding short term CO2 variations which very likely do not affect the Mauna Loa observatory and our site at the same time, and (2) to be able to better fill larger data gaps in the observatory CO2 concentration time series including the possibility to extrapolate the time series (e.g. 2019 data was not yet available at the time of data processing).

As discussed in Sect. 4 in the manuscript, there is a good agreement between the thus derived CO2 reference and the CO2 measurements at our site when the gas analyzer had been freshly calibrated (Fig. 4). At these times, the daily median of measured CO2 concentration is approximately 10-15 ppm lower than the reference. An underestimation of 15 ppm around 400 ppm means about 3.75 % error in concentration, which leads to roughly 1.5 % error in flux for CO2 (the % error in flux is roughly 40 % of the % error in concentration). Considering that the measured concentrations are often 100 to several 100 ppm away from the (assumed) real concentration and that this causes great errors in the raw flux and the WPL correction, the drift correction based on our CO2 reference can be expected to greatly reduce these errors.

[Figure]

*Fig. 4: CO2 mixing ratios during times with calibration*

However, we agree with the reviewer that using the background concentration of Mauna Loa, which is at a distance of more than 11,000 km from our site, is unfavorable. We have tested our approach with the CO2 concentration data from Mt. Waliguan (Dlugokencky et al., 2020), which is at a distance of only about 1100 km from our study site, still on the Tibetan Plateau. We agree with the reviewer, that the measurements from this site probably better reflect the local background CO2 concentration at Nam Co. Figure 5 shows that there is a good agreement between the two time series, with the one from Mt. Waliguan exhibiting a more pronounced annual variation, with earlier minimum and maximum concentrations. This is probably due to the fact that Mt. Waliguan is situated right within the continental biosphere, rather than in the middle of an ocean. In order to generate a CO2 concentration time series with a 30-minute resolution we used the same approach as for the Mauna Loa data and fitted the model discussed above to monthly averages of the weekly flask measurements from Mt. Waliguan. The drift correction was re-calculated with the new CO2 reference and all subsequent flux calculations are now based on this data. The sections in the manuscript are updated accordingly and the model for the derivation of a 30-minute reference time series from observatory data is introduced and discussed.

[Figure]

*Fig. 5: CO2 concentration time series from Mauna Loa (red, daily average in-situ samples) and Mt. Waliguan (blue, weekly flask samples) atmospheric observatories.*

**3.3 Why not use the AGC to discard fluxes with dirty windows?**

The AGC is a (loose) proxy for contamination, which is exactly what we address with the drift correction. The drift correction legitimately allows us to recover data with high AGC. Hence, we explicitly do not apply a general AGC filtering to the data set. Of course, when the AGC value rises above a certain level, the uncertainty in the signal is stronger so the overall quality of the drift correction could be degraded. We will add the AGC to the revised data set so that it may be used for further analyses.

**3.4 Why does the qc scheme not efficiently remove the 612 data points with co2_mixing_ratio close to zero?**

This problem is most probably a bug in the EddyPro software implementation. There are several arguments that hint in that direction: (1) The zeros (or close to zero) occur only in the co2_mixing_ratio. The values for co2_molar_density, co2_mole_fraction and co2_flux seem fine. (2) The corrupted co2_mixing_ratio occurs exactly every first entry after there was a data gap in the H2O reference concentration (possibly due to missing low frequency measurements). And indeed, when looking at the high frequency h2o_molar_density_li7500 we also see corrupted values every first entry after there was a data gap in the low frequency reference time series. Because the conversion from molar density to mixing ratio (of CO2 and H2O) depends on the h2o_molar_density, the corrupted values stem indeed from a software bug. The software implementation of the drift correction is still in development and was not yet officially released (i.e. you cannot use it via the GUI). A notification to fix the bug was filed and a fix will probably be implemented in one of the upcoming releases. Meanwhile, we removed erroneous CO2 and H2O mixing ratios manually by setting plausibility boundaries and added a short notice in Sect. 2.4. We are confident that our work around is appropriate

because the Li-7500 (as an open path instrument) anyway measures gas concentration in molar density first and calculates mole fraction and mixing ratio only afterwards. Hence, the calculated fluxes are not affected and can be used for analyses.

**Changes in the manuscript concerning comment 3.1:** No changes have been made in the manuscript.

**Changes in the manuscript concerning comment 3.2:** We replaced the Mauna Loa reference time series with the time series from Mt. Waliguan throughout the manuscript.

In the Methods Sect. 2.3 *Drift correction* we added information about the use of Mt. Waliguan as a reference time series and introduce the model used to derive the $CO_2$ concentration offset: "[…] we used the $CO_2$ mixing ratio measurements from Mt. Waliguan atmospheric observatory (years 2005-2018, Dlugokencky et al., 2020), situated approximately 1100 km NE of Nam Co, still on the Tibetan Plateau. We averaged the weekly flask measurements to monthly means and fitted the following model to the data:

$$CO_{2\,ref} = p_1 + p_2*t + p_3*cos(2*\pi*t/365) + p_4*sin(2*\pi*t/365) + p_5*cos(4*\pi*t/365) + p_6*sin(4*\pi*t/365),$$

where t is the decimal time in days and $p_i$ are the fit parameters. This model was used to generate the 30-minute $CO_2$ concentration reference time series. The rationale for using this model rather than a linear or spline interpolation was to mimic the general pattern of the atmospheric $CO_2$ background concentration while excluding short term $CO_2$ variations which most likely do not affect the Mt. Waliguan observatory and our site at the same time. We then calculated the $CO_2$ offset used for the drift correction on a daily basis, as the difference between the daily medians of the measured $CO_2$ concentration and the reference $CO_2$ concentration. Hence, one offset value was applied to all 30-minute $CO_2$ concentration measurements of each individual day. In contrast, the $H_2O$ offset was determined as the difference between the 30-minute $H_2O$ concentration measured by the Li-7500 gas analyzer and the $H_2O$ concentration calculated from auxiliary low frequency measurements of relative humidity, temperature and air pressure. The time series of 30 minute concentration offset values were imported as dynamic metadata file in EddyPro. Together with the sensor specific calibration information […]."

In the Sect. 4 *Discussion* we now discuss the uncertainty resulting from the use of a reference time series: "To be more precise on that, the measured daily medians remain approximately 10-15 ppm lower than the model right after user calibration was performed. An underestimation of 15 ppm around 400 ppm means about 3.75 % error in concentration, which leads to roughly 1.5 % error in flux for $CO_2$ (the % error in flux is roughly 40 % of the % error in concentration, Fratini et al., 2014). Considering that the measured concentrations are often 100 to several 100 ppm away from the (assumed) real concentration and that this causes great errors in the raw flux and the WPL correction, this correction can be assumed to greatly reduce these errors. As seen in Figs. 3 and 4, after drift correction, the mean $CO_2$ and $H_2O$ concentrations are very close to the (assumed) values. So even though not completely accurate, this strategy is expected to at least reduce the inaccuracy of the computed fluxes."

**Changes in the manuscript concerning comment 3.3:** No changes have been made in the manuscript.

**Changes in the manuscript concerning comment 3.4:** In the Results Sect. 3.1 *Drift correction* we added information about the software bug and its implications: "However, the software implementation of the drift correction is still in development and was not yet officially released (i.e. you cannot use it via the GUI), so unfortunately it still contains a software error (bug): Every data gap in the $H_2O$ reference concentration time series (due to e.g. missing low frequency meteorological data) produces a corrupted $H_2O$ mixing ratio record in the following half hour, which also affects the calculation of the

CO2 mixing ratio. This issue was raised with the EddyPro developers and will be fixed in one of the upcoming releases. Because this error does not affect the calculation of the fluxes or other variables, we removed the erroneous values by setting plausibility limits."

**H2O concentration:**

**Reviewer comment 4:** The H2O concentrations provided in the data-file are not the once from the LI7500 but the once from the biomet data, i.e. the temperature and relative humidity sensor. This might be okay for a normal data set where no issues are present with drifts, dirty windows, concentration etc. But here I would highly recommend to provide or at least look at the water vapor concentration of the LI7500. When you use EddyPro for processing I think the only way to get the true LI7500 H2O concentrations is when you run the processing without providing the biomet file. The point here is that you can't use the concentration as a quality criteria. You can actually see that by looking at the number of qc_co2_mixing_ratio_composite and qc_h20_mixing_ratio_composite. The number of bad data for qc_co2_mixing_ratio_composite (==2) is 12730 and for qc_h2o_mixing_ratio_composite (==2) is 203. If the concentration of the one is bad usually also the other one is bad. Especially when this is due to dirty windows, precipitation, snow frost, etc… I really encourage you to use the real LI7500 water vapor concentrations to select a criterion to remove bad data and also bad fluxes of h2o. In principle I would recommend to provide the raw CO2 and H2O concentrations and the corrected once.

**Response to reviewer comment 4:** We agree with the reviewer that the mean H2O concentrations of the Li-7500 should be included in the data set. The Li-7500 H2O data can be taken from the EddyPro statistics output files. In the updated data set, all CO2 and H2O concentrations involved in the processing of the data set – non-drift corrected and drift corrected CO2 and H2O high frequency, as well as H2O low frequency concentrations – are included. Furthermore, we agree with the reviewer that the H2O concentration QC flag must be derived from the corrected high frequency H2O concentration signal. This is corrected in the updated data set, so that the same QC scheme is applied for H2O (low and high frequency) as for the CO2 gas concentration data. Please note that the gas concentrations are also used during the SSH correction. For the approach of Burba et al. (2008), EddyPro uses the quality filtered H2O data from the slow meteorological probe rather than the high frequency data from the Li-7500. To be consistent, we also used H2O data from the slow meteorological probe for the SSH approach of Frank and Massman (2020). An explanatory text was added to Sect. 2.4, 2.5 and also to the data description file.

**Changes in the manuscript concerning comment 4:** In the Methods Sect. 2.4 *Quality filtering* we included the following section: "Please note that the H2O gas densities and concentrations in the EddyPro full output file are calculated mainly from low frequency measurements of air temperature, pressure and relative humidity, probably because these are deemed more accurate than the high frequency measurements of the IRGA. To enhance comparability, we extracted the high frequency H2O gas densities from the EddyPro statistics output file (variable 'mean(h2o)') and calculated the mole fraction and mixing ratio. These variables were quality filtered following the same scheme as above and are supplied with the data set (suffix: _Li7500)."

In the Methods Sect. 2.5 *Sensor self heating correction* we added the following note: "Please note that we used the quality filtered variable co2_molar_density from the EddyPro full output file in order to reproduce the calculations."

**Minor comments:**

**Reviewer Comment 5:** Please include the countries to which the southern and western part of the TP belongs. I guess Nepal, Pakistan, India and Bhutan.

**Response to reviewer comment 5:** Thanks for the suggestion, the names of neighboring countries have been included in the text.

**Changes in the manuscript concerning comment 5:** The respective sentence in the introduction was rephrased: "It has an area of about 2.5 million km$^2$ at an average elevation of > 4000 m above sea level and includes the entire southwestern Chinese provinces of Tibet and Qinghai, parts of Gansu, Yunnan, Sichuan, as well as parts of India, Nepal, Bhutan and Pakistan."

**Reviewer Comment 6:** The sine-cosine model is not explained. Why not directly using a spline function or even a moving average? Did you use the flask samples or the continuous? The pattern of the CO2 concentrations is not really a sine or cosine.

**Response to reviewer comment 6:** We explained the model and the rationale behind it under the point 3.2. above.

**Changes in the manuscript concerning comment 6:** Please see point "Changes in the manuscript concerning comment 3.2" above.

**Reviewer Comment 7:** I think the u* filtering should be applied. Just because there s wind does not mean there is no relationship. For grassland values around u* values of 0.1 m/s are not so uncommon and that accounts in your data set for 15% of all u* values. The red line in the plot shows the cumulated density function multiplied by 3000 to fit the scale. Green vertical line is at u*== 0.1 m/s.

**Response to reviewer comment 7:** We agree with the reviewer and added u* filtering to our processing pipeline. We think that one overall u* threshold is sufficient because vegetation height remains very low throughout the year. The threshold was estimated using REddyProc. We added a quality flag (qc_ustar), indicating whether a 30-min flux has a lower u* than the estimated threshold. These were then excluded from further analyses. A short paragraph was added to the text and in the data description file.

**Changes in the manuscript concerning comment 7:** We added the following text to Sect. 2.4 *Quality filtering*: "To identify periods with insufficient turbulent mixing, we estimated the friction velocity (u*) threshold using the REddyProc R package. Fluxes with u* < 0.08 m s$^{-1}$ were excluded from subsequent calculations."

Furthermore, Table A1 now includes the proportion of available fluxes after u* filtering and a quality flag (qc_ustar) was added to the data set and to the data description file.

**Reviewer Comment 8:** Formular 7 and 8 you use mu and sigma which are usually the population mean and its standard deviation. I know you took it from the paper of Burba but there are plenty of other variables one can use. For the uncertainty analysis of the WPL I have only a gut feeling that this is wrong but it would be good if you would get some statisticians input and explain why this is valid to

do. In principle each value n the formula has an uncertainty e.g. Ta which propagates in Cp and rho. Sorry for not being more helpful on this one. But generally, for the overall uncertainty I would rather take the NEE_fsd to calculate the uncertainty of the fluxes. It is including not only the random error but also temporal variability and spatial heterogeneity. There is a paper comparing these uncertainties with each other I think in a savanna (sorry I can't recall the author maybe worth a look).

**Response to reviewer comment 8:** We revised the error propagation section. We now agree with the reviewer that that the error propagation calculation is not adequate, mostly due to the fact that the errors in the fluxes involved (H, LE, CO2) are not statistically independent of each other. Hence the standard error propagation method cannot be applied and has to be discarded. In the updated data set, we rely on the random flux uncertainties calculated by EddyPro following Finkelstein and Sims (2001). These should sufficiently cover the flux uncertainty that originates due to the sampling errors.

Additionally, the standard deviation of the marginal distribution sampling (MDS) gap-filling procedure (Reichstein et al., 2005) can be used as a measure for flux uncertainty. In the "savanna" paper mentioned by the reviewer, El-Madany et al. (2018), compared the random flux uncertainty (Finkelstein and Sims, 2001), the standard deviation of the marginal distribution sampling (MDS) gap-filling procedure (Reichstein et al., 2005), and a Two-Tower uncertainty (Hollinger and Richardson, 2005) from three different towers in the same ecosystem (but with non-overlapping footprints). As we only have one flux tower, we cannot apply the two-tower uncertainty but the standard deviation of the MDS gap-filling (NEE_fsd) is included in the data set. The NEE values for gap filling are (mostly) calculated using a look-up table approach with quite narrow meteorological bins (bin width: Rg = 5 W m$^{-2}$, VPD = 5 hPa, Tair = 2.5 °C) within a short time window (+- 7 days). During these conditions, the vegetation should not change a lot and hence, the ecosystem response to atmospheric drivers should be the same. Any variability of the flux measurements probably stems from the temporal (+- 7 days sampling window) and spatial (changes in footprints between 30-min fluxes) heterogeneity at the site. Hence, we could use the NEE_fsd as an additional measure for flux uncertainty to complement the random uncertainty estimation of Finkelstein and Sims (2001). As expected, the random uncertainty remains much lower than the NEE_fsd (medians of 0.2 and 0.5 µmol m$^{-2}$ s$^{-1}$, respectively, Fig. 6).

**Changes in the manuscript concerning comment 8:** The random flux error propagation via the WPL correction was removed throughout the manuscript.

In Sect. 2.7 *Flux uncertainty estimation* we added the following text describing the rationale behind the use of NEE_fsd as a flux uncertainty estimate: "Because the RE method is based on the half hourly auto- and cross-correlation of the vertical wind component and the scalar of interest (e.g. air temperature or gas concentration), it contains only very limited information about spatial, temporal or meteorological variability (El-Madany et al., 2018). During the MDS gap filling procedure (Reichstein et al., 2005), the missing NEE values are (mostly) calculated using a look-up table approach with quite narrow meteorological bins (bin width: Rg = 5 W m$^{-2}$, VPD = 5 hPa, Tair = 2.5 °C) within a short time window (+/- 7 days). During these conditions, the vegetation should not change a lot and hence, the ecosystem response to atmospheric drivers should remain the same. Any variability of the flux measurements probably stems from the temporal (+/- 7 days sampling window) and spatial (changes in footprints between 30-min fluxes) heterogeneity at the site. Hence, we could use the standard deviation of the fluxes used for gap filling (NEE_fsd) as an additional measure for flux uncertainty to complement the random uncertainty estimation of Finkelstein and Sims (2001)"

We added an additional Fig. 8 which shows the monthly mean diurnal course and the annual course of the daily mean of the CO2 flux and uncertainty estimates (RE and NEE_fsd):

[Figure]

*Fig. 6: The new Fig. 8 in the manuscript: Monthly mean diurnal course and annual course of daily mean of the CO2 flux and uncertainty estimates: RE is the random uncertainty following Finkelstein and Sims (2001), NEE_fsd is the standard deviation of values used for gap filling after Reichstein et al. (2005).*

Furthermore, we added a new Results Section 3.4 *Flux uncertainty estimation* describing the results of the uncertainty estimation and describing the new Fig. 8: "Figure 8 shows the mean diurnal and annual cycle of the CO2 flux and the respective uncertainties. The two uncertainty estimates (RE, NEE_fsd) follow a distinct distribution, thereby reflecting the different sources of error they represent. As expected, the random uncertainty remains much lower than the standard deviation of the gap-filled fluxes (medians of 0.2 and 0.5 µmol m$^{-2}$ s$^{-1}$, respectively). The RE exhibits roughly the same magnitude throughout the year whereas the NEE_fsd increases with increasing flux magnitude. Concerning the diurnal course, we see lower uncertainties during nighttime and winter than during daytime and summer. The RE is generally smaller during night while during daytime, the uncertainties almost converge."

In the Sect. 4 *Discussion* a sentence was added to describe which uncertainties are covered by our estimates: "The random uncertainty estimates described above represent random flux components as well as spatial heterogeneity, temporal variability, and small meteorological variability while neglecting other sources of random flux errors such as instrument noise."

[Figure]

*Fig. 7: Histograms of two different CO2 flux uncertainties. MDS is the standard deviation of the gap-filling procedure after Reichstein et al. (2005) and RE is the random uncertainty estimate following Finkelstein and Sims (2001).*

**Reviewer Comment 9:** The sentence in line 425 "The wind direction distributions of wind speed and TKE, as well as the analysis of cumulative footprints suggest that the several buildings which were constructed in close vicinity of the tower do exert some influence on the flow regime while not violating basic EC assumptions. Nevertheless, fluxes originating mainly from the disturbed areas should be excluded from further analyses as they may be compromised by human activities." You have all indications that the flow was clearly disturbed and you still conclude that the assumptions of eddy covariance are met? How does that go together?

**Response to reviewer comment 9:** The statement, that basic EC assumptions are not violated was made with respect to the fulfillment of SS and ITC tests. Basically meaning, that the values are not usable, although qc_co2_flux == 0 and should hence be excluded from further analyses, as we state correctly in the subsequent sentence. We agree that the sentence is unclear so we changed the formulation to avoid confusion about this.

**Changes in the manuscript concerning comment 9:** The respective sentence in the Sect. 5 *Conclusions* was changed: "The wind direction distributions of wind speed and TKE suggest that the several buildings which were constructed in close vicinity of the tower do exert some influence on the flow regime. Fluxes originating from the disturbed areas should be excluded from further analyses as they may be compromised by human activities."

**Reviewer Comment 10:** For the data description file some more info on the uncertainties would be great. The differences are not directly clear to the reader.

**Response to reviewer comment 10:** The variables "rand_err_co2_flux_wpl" and "rand_err_h2o_flux_wpl" are not calculated anymore so they had to be removed from the data description file. Concerning "rand_err_co2_flux" and "rand_err_h2o_flux", information about the calculation method was added to the data description file.

**Changes in the manuscript concerning comment 10:** No changes have been made in the manuscript. In the data description file we added a short statement after which method (Finkelstein and Sims, 2001) the random error was calculated.

**Additional changes made by the authors:**

We added the DOI and source of the second data set repository (National Tibetan Plateau Data Center) in the abstract.

In the Methods Sect. 2.4 *Quality filtering* we stated that we filtered for discontinuities while in fact we did not use this measure. We corrected the formulation accordingly.

During gap filling, we additionally filtered for night time fluxes < -0.1 µmol m$^{-2}$ s$^{-1}$ and excluded the upper and lower 0.2 % in order to exclude implausible fluxes from the gap-filling procedure. We added a note in the Methods Sect 2.6 *Gap filling*: "Prior to the processing, we excluded the lower and upper 0.2 % of the fluxes and discarded physiological implausible night time fluxes < -0.1 µmol m$^{-2}$ s$^{-1}$."

We introduced the following figure showing the offset between the Li-7500 H2O concentration measurements and the low frequency reference time series before and after drift correction. This should help to illustrate the effect of the drift correction procedure.

[Figure]

*Figure 8: The new Fig. 4. In the manuscript: Half hourly H2O dry air mixing ratios and low frequency reference concentration before and after drift correction. H2O mixing ratios have been checked for repeating values and outliers using the same algorithms as in Sect. 2.4. Please note the different y-axis scales.*

[revised manuscript text omitted]

**2.5**

~~In order to conduct meaningful estimations of the fluxes, the area "seen" by the measurement should represent the ecosystem of interest and the flow regime should be as undisturbed as possible (Aubinet et al., 2012). During the long measuring period, spanning nearly 15 years, several buildings and scientific infrastructure were constructed in close vicinity of the eddy covariance tower. During the development of the NAMORS, from the foundation with only a few tents in 2005 to a well-equipped research station in 2019, we approximated five times with significant changes in constructions (2). In 2009 the PBL container, the shed and the solar panel were set up. In 2010 the main building and the green house were constructed. In 2012 the shed was rotated to become the laboratory and the tool shed next to the greenhouse was added. Finally, in 2019 the garage was relocated and extended south of the laboratory and the solar panels were removed. To account for possibly disturbed turbulence, we applied the planar fit axis rotation for three different wind sectors during flux calculation (see Sect. 2.2). To assess possible influences on the flow and turbulence regime, we analyzed the wind direction distribution of the mean wind speed and the turbulent kinetic energy. Furthermore, we estimated the source area of the flux measurements by calculating cumulative footprints using the model of Kormann and Meixner (2001). This footprint model was developed for non-neutral atmospheric conditions, therefore all measurements were checked for the stability parameter $(z-d)/L \neq 0$. The model relates a flux to a certain direction in a certain distance around the EC station, depending on the measurement height, wind direction and wind speed, friction velocity, atmospheric stability and the cross stream wind component. The cummulative footprints (Fig. ??) were calculated and plotted using FREddyPro R package. However, EddyPro also allows for the computation of flux footprints using Kormann and Meixner (2001) but supplies only the distances of flux contributions (~~

**Table 2.** Disturbed wind sectors and times.

| From | To | Back of CSAT3 | PBL container | Main buildings |
|------|-----|--------------|---------------|----------------|
| 2005-12-04 | 2009-06-30 | 305°–325° | - | 260°–280° |
| 2009-06-30 | 2010-01-30 | 10°–30° | 30°–50° | 260°–280° |
| 2010-01-30 | 2011-12-31 | 10°–30° | 30°–50° | 250°–300° |
| 2012-01-01 | 2018-12-31 | 10°–30° | 30°–50° | 250°–315° |
| 2019-01-01 | 2019-09-07 | 10°–30° | 30°–50° | 245°–315° |

**2.5 Sensor self heating correction**

When using an open path IRGA, it is necessary to correct for air density fluctuations caused by fluctuations of temperature and water vapor in the measurement path. The WPL correction compensates for the naturally occurring density fluctuations and should be applied in any case (Webb et al., 1980). Furthermore, especially during cold conditions (low temperatures below -10 °C), an apparent $CO_2$ uptake may be measured, which is caused by conductive, convective, and radiative heat exchange processes happening in the measurement path (Burba et al., 2008). These stem from heating of internal electronics during normal operation, as well as solar radiation encountered by different instrument parts surrounding the open sampling path of the sensor. This correction is necessary for pre-2010 models of the Li-7500 or for newer instruments (e.g. Li-7500A, Li-7500RS) with summer setting but used in a very cold environment (Oechel et al., 2014). Although the size of the heating correction is quite small (i.e. 10-50 times smaller than the WPL-correction) the small bias can lead to an overestimation of net ecosystem $CO_2$ uptake when integrating over long periods in cold environments (Oechel et al., 2014). Burba et al. (2008) developed a correction procedure which is well tested and fully implemented in the EddyPro software. The procedure depends on a range of correction factors, which were developed from the original sensor setup in Nebraska, USA. Our study site on the Tibetan Plateau displays different environmental conditions than those in which the correction was tested, namely an inclined IRGA, lower ambient temperatures and strong winds, as well as possible snow and ice deposits on parts of the instrument. ~~No independent gas concentration measurements are available from our study site which could be used for fine tuning of the air density correction. Hence, we applied the approach by Oechel et al. (2014), who developed an empirical method to calibrate the correction of EC measurements, originally for a sensor in Alaska. We made the same assumptions as Oechel et al. (2014) concerning the relationship between the vertically aligned sensor Burba et al. (2008) used in the original method in Nebraska in comparison to our aligned sensor in Tibet: The sensor consists of a bottom, cylinder-shaped part and a top, ball-shaped part which are differently exposed to ambient conditions (e.g. solar radiation) depending on the inclination of the sensor. While the top, ball-shaped part of the sensor is about equally exposed in Nebraska and Tibet, we assume the inclined bottom cylinder in Tibet to be more exposed to radiation than the vertical aligned bottom cylinder in Nebraska. Its temperature ($T_{botTI}$) is a combination~~ In a recent publication, Frank and Massman (2020) tested the correction procedure for a "cold, windy, high-elevation mountainous site" and found inconsistencies in the Burba et al. (2008) correction: (1) The

240 Burba et al. (2008) correction contains boundary layer adjustment terms for non-flat surfaces but the top- and bottom surfaces of the Li-7500 are flat. This leads to an underestimation of the surface heat fluxes which is an order of magnitude too small. (2) The weightings

$$T_{botTI} = xT_{botNE} + (1-x)T_{topNE}$$

245 ~~The weighting factor $x$ has to be parameterized by calculating the sensor self heating correction using Method 4 (submethod: linear regression with air temperature Burba et al., 2008) for multiple weighting factors. The optimal weighting factor is selected on the basis of two criteria: First, periods have to be identified when a change in $CO_2$ efflux with temperature can be assumed to be negligible. For our study site on the Tibetan Plateau, we assume negligible $CO_2$ efflux when air temperature is below -15 °C and soil moisture below 2 % during the months of January and February. The ground can be~~
250 ~~regarded as continuously frozen since two months, with mean air temperatures below 0 °C at least since November. By then, freezing should have pushed any excess $CO_2$ out of the soil. During these cold conditions, the flux-to-air temperature slope was calculated for every weighting factor (0-100). The $CO_2$ fluxes should not change with temperature during these cold conditions, hence the closer the slope to zero,~~ and top surface heat fluxes are "improbable and an order of magnitude too large" (Frank and Massman, 2020). While these errors canceled out during the study of Frank and Massman (2020), this may not be
255 the case for other field sites. Following the recommendations of Frank and Massman (2020), we discarded the boundary layer adjustment terms for non-flat surfaces from the calculations and applied their newly calculated weightings of the bottom and top surface heat fluxes, thus emphasizing the role of
260 ~~would implicate, that the ambient air is warmer than the sensor surface which is implausible at temperatures below 0 °C. Hence, the number of negative daily corrections at air temperatures below 0 °C should be small, which is the second criterion to find the optimal weighting factor. The optimal weighting factor is then used to correct all measurements when ambient temperature is below 0 °C. We used a radiation threshold of 5 W m$^{-2}$ to distinguish between daytime and nighttime. The method is described in detail in Oechel et al. (2014) and the R script with the exact calculations can be found in the supplementary material~~spar
265 in self heating. We first reproduced the Burba et al. (2008) correction like it is implemented in EddyPro and then adjusted the equations as described above. Please note that we used the quality filtered variable co2_molar_density from the EddyPro full output file in order to reproduce the calculations.

**2.6 Gap filling**

In order to obtain a $CO_2$ flux time series as complete as possible, we filled the data gaps using the marginal distribution
270 sampling (MDS) algorithm (Falge et al., 2001; Reichstein et al., 2005), implemented in the REddyProc R package by Wutzler et al. (2018). Depending on the length of the data gap and the availability of the meteorological input variables radiation (Rg), air temperature (Tair) and water vapor deficit (VPD), the missing $CO_2$ flux values are derived from a look up table (LUT)

or from mean diurnal course (MDC). The LUT approach replaces the missing value with the average value under similar meteorological conditions within a certain time window. Meteorological conditions are similar if Rg deviates not further than 50 W m$^{-2}$, Tair not further than 2.5 °C and VPD not further than 5.0 hPa. If no similar conditions can be found within an appropriate time window, the missing value is replaced using the average value at the same time of the day (1 hour) (MDC). If the missing value can not be filled during the initial time period (7 - 14 days), the time window size is increased and the procedure repeated until the value can be filled or the data gap gets too long for reliable gap filling (i.e. > 60 days). As horizontal wind speeds are generally very high (lowest percentile = 0.47 m s$^{-1}$) at our study site, we did not filter for low friction velocity. The full MDS algorithm is described in Wutzler et al. (2018) and the R script used in this study can be found in the appendix.

Prior to the processing, we excluded the lower and upper 0.2 % of the fluxes and discarded physiological implausible night time fluxes < -0.1 μmol m$^{-2}$ s$^{-1}$. To estimate the uncertainty of the gap-filling gap filling procedure, we used the method implemented in REddyProc R package, which, besides filling real gaps, creates artificial gaps from otherwise available data and fills them in the same way as if it was a real gap (see section 2.6). The model-value residual should be considered when aggregating the gap-filled time series to daily or annual estimates of NEE, GPP and R$_{eco}$. We included the filled values for the artificially created data gaps, as well as quality flags for the gap filling procedure, with "0" for measured data, "1" for high reliability, "2" for intermediate reliability and "3" for poor reliability of the gap-filled values. The full MDS algorithm is described in Wutzler et al. (2018) and the R script used in this study can be found in the appendix.

**2.7 Flux uncertainty estimation**

As with all measurements, the reported fluxes are subject to uncertainty, consisting of a systematic and a random part. Systematic uncertainties may occur e.g. from having an imperfect measurement setup or, like in our case, due to limited maintenance and calibration of the sensors (see section 2.3). We applied a wide range of methods to filter and compensate for systematic errors. Most importantly, we tested for fulfillment of basic EC assumptions using integral turbulence characteristics and steady state test (e.g., Foken and Wichura, 1996) and compensated for air density fluctuations and high- and low-frequency losses (see Sect. 2.2, 2.4 and 2.5). In contrast to systematic uncertainties, random errors do not bias the flux in any direction but reduce the overall confidence (i.e. precision) of the reported values (Richardson et al., 2012). Random uncertainties mainly arise from the stochastic nature of turbulence, footprint variability, as well as from instrument noise and the resolution at which samples are recorded (Richardson et al., 2012). Hence, it is important to estimate the random uncertainty, especially in places with rather low magnitude of fluxes, as it is in our case. We estimated the random flux error (RE) using the mathematically rigorous and fully implemented approach by Finkelstein and Sims (2001). This method calculates the random flux errors uncertainty arising from insufficient sampling of large eddies with high spectral energy, the so-called sampling error. As these large turbulences appear irregularly during sampling, the error is random and can be estimated. First, the so-called Integral Turbulence time-Scale (ITS) is calculated. Basically, the ITS is the covariance between vertical wind velocity and gas concentration as a function of lag time between these two time series (Holl et al., 2019). With increasing time lag, the cross correlation function typically decreases towards values close to zero, indicating an increasing non-correlation of the two time series. In practice, the correlation function must be stopped, otherwise it would go infinitely towards zero. We stopped the integral as soon as the

cross-correlation function (– which always starts at 1 )– crosses the x-axis (i.e. first crossing 0). In case the cross-correlation function would never cross the x axis, a default time value can be provided at which the function is stopped. We set this "maximum correlation period" to 5 s in order to keep computational performance high. Once the ITS is calculated, the random uncertainty of the fluxes RE can be estimated based on the calculation of the "variance of covariance" (Finkelstein and Sims, 2001).

It is important to note, that the random error estimate applies to the uncorrected fluxes, i.e. before correction for spectral attenuation and air density fluctuations. In order to estimate the random error of the finally corrected fluxes, the propagation of error through the corrections has to be taken into account, especially for the estimation of long-term NEE (Liu et al., 2006) . First, we multiplied the errors with the same spectral scaling factor as the fluxes to account for spectral attenuation. Then, we used Eq. (1) and (2) from Burba et al. (2008) to apply the WPL correction to the flux errors. Following basic concepts of error propagation, we corrected the random uncertainties the same way as if they were fluxes but adding them up in rooted quadrature using Eq. (??) for random error of water vapor flux and Eq. (??) for random error of $CO_2$ flux.

$$RE_E = (1 + \mu\sigma)\sqrt{RE_{E_0}{}^2 + (\frac{RE_H}{\rho C_p}\frac{\rho_c}{T_a})^2}$$

$$RE_{FC} = \sqrt{RE_{FC_0}{}^2 + (\mu\frac{RE_E}{\rho_d}\frac{\rho_c}{1+\mu\sigma})^2 + (\frac{RE_H}{\rho C_p}\frac{\rho_c}{T_a})^2}$$

$RE_E$ and $RE_{FC}$ are the WPL-corrected water vapor and $CO_2$ flux errors (kg m$^{-2}$ s$^{-1}$) . $RE_{E_0}$ and $RE_{FC_0}$ are the initial water vapor and $CO_2$ flux errors (kg m$^{-2}$ s$^{-1}$), not corrected for WPL, but already multiplied by the spectral correction factors. $RE_H$ is the sensible heat flux error, already corrected for WPL and spectral correction factor (W m$^{-2}$). $\mu$ is the ratio of molar masses of air to water ($\mu = 1.6077$), $\sigma$ is the ratio of the mean water vapor density ($\rho_v$ in kg m$^{-2}$ s$^{-1}$) to mean dry air density ($\rho_d$ in kg m$^{-2}$ s$^{-1}$) . $\rho_c$ is the mean ambient $CO_2$ density (kg m$^{-2}$ s$^{-1}$) and $\rho$ is the mean total air mass density (kg m$^{-2}$ s$^{-1}$) . $C_p$ is the air heat capacity (J kg$^{-1}$ K$^{-1}$). The a posteriori sensor self heating correction (Sect. 2.5) does not include other scalars with quantified random error within its equations. Hence, the propagated random $CO_2$ flux error remains the same before and after self heating correction. 
[revised manuscript text omitted]
. ~~The cumulative footprints show complementary behavior. The main source area is a 150 m circular around the EC station, covering the buildings and sealed area, as well as the alpine steppe ecosystem within and outside the fenced area. In general, the size of the footprint gets smaller the stronger the atmosphere is mixed. The footprints at NAMORS follow this scheme, indicating an increase of the turbulent mixing in the lee of the different buildings. Fluxes $\geq$ 50 % contribution from the disturbed areas were excluded from further analyses. The ROI boundary is indicated in Fig. 2.~~

400

405 **3.3 Sensor self heating correction**

[Figure]

**Figure 6.** Wind roses showing the wind speed distribution and turbulent kinetic energy (TKE) in 5 ° binned wind directions at NAMORS EC station before and after 2010.

 The monthly mean diurnal course of the three $CO_2$ flux time series in Fig. 7 clearly shows the effect of the sensor self heating correction  during cold conditions (air temperature < 0 °C)

410   . The effect of the correction procedure following Burba et al. (2008) and the revised equations of Frank and Massman (2020) are very similar. We see various problems associated with the SSH corrections: (1) the corrections create strong artifacts during the transition between day and night, (2) SSH-corrected nighttime $CO_2$ fluxes

415   ~~steeper $CO_2$ flux-to-air temperature slope, which is physiologically unlikely. Decreasing the weighting factor led to a higher number of negative daily corrections which is implausible from the fundamental thermal exchange between the instrument which is controlled at +30 °C and ambient air temperatures below 0 °C. Figure 8 shows that the apparent $CO_2$ uptake during cold conditions was efficiently removed by the correction. The heating correction had the greatest effect during daytimewhen solar radiation additionally heats the inclined sensor. Before the correction,~~ 
[revised manuscript text omitted]

765     https://doi.org/10.1016/j.agrformet.2014.12.013, 2015.